# Ginseng-Derived Carbon Quantum Dots Enhance Systemic Exposure of Bioactive Ginsenosides and Amplify Energy Metabolism in Mice

**DOI:** 10.3390/pharmaceutics17111485

**Published:** 2025-11-17

**Authors:** Huiqiang Liu, Xin Sun, Bo Yang, Chuan Lin, Xiwu Zhang, Hui Sun, Xiangcai Meng, Yufeng Bai, Tao Zhang, Guangli Yan, Ying Han, Xijun Wang

**Affiliations:** 1State Key Laboratory of Integration and Innovation of Classical formula and Modern Chinese Medicines, National Chinmedomics Research Center, National TCM Key Laboratory of Serum Pharmacochemistry, Metabolomics Laboratory, Department of Pharmaceutical Analysis, Heilongjiang University of Chinese Medicine, Heping Road 24, Harbin 150040, China; huiqianglliu@sina.com (H.L.);; 2State Key Laboratory of Quality Research in Chinese Medicine, Macau University of Science and Technology, Macau 999078, China; 3Harbin Chengcheng Institute of Life and Material, Harbin 150040, China

**Keywords:** ginseng, carbon quantum dots, energy metabolism, in vivo active components, drug metabolism

## Abstract

**Objective:** To overcome the extremely low oral bioavailability of ginsenosides in traditional ginseng preparations, this study aimed to evaluate the efficacy of a novel ginseng-derived carbon quantum dots (G-CQDs) delivery system and to elucidate its core bioactive constituents and integrated mechanisms of action. **Methods:** G-CQDs were prepared from ginseng roots via ultrahigh-speed nitrogen jet pulverization combined with far-infrared pulse-assisted hydrothermal carbonization. Their physicochemical properties were characterized by transmission electron microscopy, Fourier-transform infrared spectroscopy, and fluorescence spectroscopy. The in vivo effects of G-CQDs versus traditional ginseng aqueous extract (G-AE) were compared in C57BL/6 mice (n = 12/group) using the PRO-MRRM-8 Comprehensive Laboratory Animal Monitoring System for real-time, non-invasive phenotyping of energy metabolism parameters (respiratory quotient, heat production, and oxygen consumption). Systemic exposure to ginseng bioactives was profiled using UHPLC-Q/Orbitrap/LTQ high-resolution mass spectrometry, followed by bivariate correlation analysis to identify key bioactive components linked to efficacy. **Results:** Compared with G-AE, G-CQDs significantly enhanced whole-body energy metabolism—respiratory quotient +2.8%, heat production +6.7%, and locomotor activity +22.9% (*p* < 0.05). A total of 110 in vitro constituents, 35 blood prototypes, and 29 metabolites were identified. Correlation analysis revealed eight core bioactive clusters linked to the metabolic benefits; all showed higher systemic exposure with G-CQDs (range +9.2% to +265.8%), notably ginsenoside Re +69.6%, cinnamic acid + O + SO_3_ +157.4%, and linolenic acid–GSH conjugate +265.8%. **Conclusions:** Carbon quantum dot technology significantly enhances the systemic exposure of ginseng bioactivities by improving solubility and enhancing gastrointestinal absorption, providing a molecular basis for its superior efficacy in regulating energy metabolism compared to conventional extracts. This study establishes a novel framework for developing high-value, bioavailability-enhanced nano-preparations from traditional medicines.

## 1. Introduction

Panax ginseng C.A. Mey. (Araliaceae), derived from the dried roots and rhizomes of the ginseng plant, has been employed for millennia across China and numerous Asian nations for disease prevention and treatment [1]. Modern pharmacological studies have revealed a broad spectrum of bioactivities, including anti-aging, anti-tumor, anti-inflammatory, antioxidant, anti-melanogenic, neuroprotective, and immunomodulatory effects [2]. However, traditional ginseng preparations face a significant bioavailability bottleneck: the prototype ginsenosides exhibit high molecular weights, pronounced polarity, and predominantly amphiphobic properties, which result in gastrointestinal absorption rates of less than 5% [3]. Moreover, they are highly susceptible to acid-catalyzed hydrolytic degradation in the acidic environment of the stomach (e.g., by hydrochloric acid), leading to significant structural transformation and loss of efficacy before reaching systemic circulation [4]. This bioavailability bottleneck not only causes substantial resource wastage, but also represents a major scientific barrier to the modern therapeutic application of ginseng.

To overcome this delivery limitation, various advanced strategies have been explored. Among them, nanotechnology-based delivery systems have emerged as a transformative approach [5,6,7]. For instance, liposomes, polymeric nanoparticles, and micelles have been investigated to enhance the solubility, stability, and intestinal permeability of poorly bioavailable phytochemicals. While promising, the search for more efficient, biocompatible, and easily scalable nano-carriers continues. In this context, carbon quantum dots (CQDs) have recently gained significant attention. CQDs are quasi-spherical carbon nanomaterials (<10 nm) characterized by surface-rich oxygen-containing functional groups (e.g., carboxyl and hydroxyl), which confer exceptional aqueous dispersibility, chemical modifiability, and presumed biocompatibility [8]. Unlike other nanocarriers, CQDs can be synthesized from natural precursors, offering a green and sustainable platform. Therefore, the CQDs strategy was chosen for this research based on their unique combination of optimal nano-carrier properties and their potential for seamless integration with herbal matrices, potentially offering a superior delivery vehicle for ginseng bioactives compared to conventional formulations or more complex synthetic nanocarriers. The novel hydrothermal method developed in this study capitalizes on these properties—transforming ginseng roots into a CQD colloidal solution through ultrahigh-speed pulverization combined with far-infrared-assisted carbonization. This process is designed to encapsulate inherently insoluble ginsenosides within a nanocarbon framework, producing a water-dispersible colloid [9], thereby effectively circumventing the solubility limitations of traditional extracts [10].

Energy metabolism imbalance serves as a fundamental pathophysiological basis for various disorders, including obesity, type 2 diabetes, and non-alcoholic fatty liver disease [11,12,13], and targeting metabolism has emerged as a promising therapeutic strategy [14]. It was selected as the central evaluation metric in this study because it provides a holistic, real-time readout of systemic physiological status and therapeutic efficacy [15]. The C57BL/6 mouse model was selected for this investigation because it is a well-established and widely used preclinical model for studying energy metabolism and metabolic diseases, ensuring that our findings are relevant and comparable to the existing literature. Recent evidence demonstrates that ginsenosides regulate energy metabolism via activation of the AMPK/PGC-1α axis and augmentation of mitochondrial function [16]. However, these insights primarily derive from cellular or ex vivo investigations, which cannot fully replicate the dynamic, real-time metabolic processes occurring in intact living organisms. To capture the dynamic nature of metabolism, this study utilizes the PRO-MRRM-8 Comprehensive Lab Animal Monitoring System, which enables continuous, simultaneous measurement of parameters such as respiratory quotient (RQ), oxygen consumption (VO_2-_), and heat production (HP) in a non-invasive manner, overcoming the limitations of static or ex vivo models. Notably, emerging evidence suggests that chiral CQDs may selectively enhance cellular glycolysis, pointing to an intrinsic energy-modulating property of nanocarbon materials [17]. Therefore, energy phenotyping serves not only to validate the improved delivery efficiency of ginseng-based CQDs but also to yield insights into their biological mechanisms. The central hypothesis of this study was that transforming ginseng into ginseng-derived carbon quantum dots (G-CQDs) would significantly enhance the systemic exposure of its bioactive components through nano-carrier effects, which would, in turn, lead to superior efficacy in regulating whole-body energy metabolism compared to a traditional ginseng aqueous extract (G-AE). This study delivers a comprehensive mechanistic account of the G-CQDs platform, uniting a novel synthesis and characterization of G-CQDs with real-time in vivo energy-metabolism phenotyping in mice to verify their superior bioactivity, and with integrated chemical analysis and correlation mapping to pinpoint the core bioactive component clusters. To our knowledge, this study provides the first in vivo energy-phenotyping evidence comparing G-CQDs with conventional G-AE, while integrating UHPLC-HRMS exposomics to connect circulating prototypes/metabolites with metabolic readouts [14]. We further show that CQD nano-carrier effects markedly elevate the systemic exposure of ginsenoside prototypes and their phase II conjugates, delineating exposure–effect clusters that plausibly drive whole-body energy metabolism.

## 2. Methods

### 2.1. Reagents and Materials

HPLC-grade methanol and acetonitrile were purchased from Thermo Fisher Scientific (Waltham, MA, USA). Formic acid was obtained from Sigma-Aldrich (Shanghai, China). Distilled water was supplied by Watsons (Guangzhou, China). Ginseng (Panax ginseng C.A. Mey.) roots, cultivated in the Jilin Province of China, were procured from Beijing Tongrentang Pharmacopoeia (Batch No: 2302063). This region is a well-known and geographically indicated production area for high-quality ginseng.

### 2.2. Preparation of G-CQDs

Ginseng roots were subjected to ultrahigh-speed impact comminution (600 m/s nitrogen gas flow velocity) for 2 impacts, yielding ultrafine powder. The powder was mixed with water at a mass ratio of 1:10 and treated in a photoelectric carbonization apparatus under far-infrared irradiation (30 pulses per second) at 120 °C for 180 s until complete carbonization was achieved. The resulting material was dispersed in water to form a colloidal solution of CQDs. The dispersion was then centrifuged at 13,000 rpm to remove insoluble residues, and the supernatant was dialyzed using a 1000 Da molecular weight cut-off membrane against deionized water for 72 h with periodic water changes to obtain purified G-CQDs. The G-CQD solution was adjusted to a final concentration of 0.15 g/95 mL for subsequent use.

### 2.3. Preparation of G-AE

Ginseng roots (30 g) were coarsely pulverized and immersed in 30 volumes of distilled water for 30 min, followed by reflux extraction at 100 °C for 1 h. The extract was first filtered through a 100-mesh filter cloth, and the residue was subjected to two additional extractions, each with 20 volumes of water under the same conditions. The combined filtrates were then centrifuged at 13,000 rpm to remove particulate impurities, and the supernatant was collected. The supernatant was concentrated to yield a G-AE with a final concentration of 0.15 g/95 mL.

### 2.4. Structural and Surface Characterization of G-CQDs

Transmission electron microscopy (TEM) and high-resolution TEM (HRTEM) imaging were performed using a JEM-F200 instrument (JEOL, Tokyo, Japan) operated at an accelerating voltage of 200 kV. The morphology, particle size distribution, and microstructure of G-CQDs were characterized by TEM (JEM-F200; JEOL, Tokyo, Japan) operated at 200 kV. Atomic lattice fringes and local ordering were examined by high-resolution transmission electron microscopy (HRTEM), and 2D-FFT of selected ROIs was used to determine d-spacings. FTIR (Fourier-transform infrared) spectra of G-CQDs were recorded using an ATR-FTIR instrument (Nicolet iS50; Thermo Fisher Scientific; Waltham, MA, USA) within 400–4000 cm^−^^1^ to identify surface functional groups. Steady-state emission spectra were collected on a fluorescence spectrophotometer (F-4500; Hitachi High-Tech; Tokyo, Japan) using a standard 1 cm quartz cuvette under UV excitation (λ_ex = 365 nm), with emission scanned across the visible range. Zeta-potential was determined on a dynamic light scattering instrument (Zetasizer Nano ZS; Malvern Panalytical; Malvern, UK) at 25 °C using aqueous dispersions of CDs, following the manufacturer’s standard protocol for electrophoretic measurements. Spectral properties of CDs were studied using ultraviolet–visible light (UV–Vis) (CECIL, Cambridge, UK).

### 2.5. In Vivo Experimental Design

A total of 36 specific pathogen-free (SPF)-grade C57BL/6 mice (weight range: 20–22 g) were obtained from Changchun Changsheng Biotechnology Co., Ltd. (License No. SCXK(Liao)2023-0001). A sample size of n = 12 per group was determined to provide sufficient statistical power based on preliminary data and consistent with established protocols for metabolic phenotyping studies in rodents, ensuring the detection of significant inter-group differences. Inclusion and Exclusion Criteria: All mice were healthy and acclimatized without incident. Pre-defined exclusion criteria included signs of severe distress, infection, or a body weight loss exceeding 15% of the initial weight; however, no animals met these criteria during the study, and, thus, all 36 mice were included in the final analysis. The mice were housed under controlled environmental conditions (temperature: 24 ± 2 °C; humidity: 50 ± 5%; 12 h light/dark cycle) for one week to allow for acclimatization. All experimental procedures were approved by the Animal Ethics Committee of Heilongjiang University of Chinese Medicine (License No. 2023113004) and conducted in accordance with ethical standards for animal research. The mice were randomly assigned to one of the following three experimental groups (n = 12/group): a control group, a G-AE group, and a G-CQDs group. Mice in the G-AE and G-CQDs groups were orally administered their respective treatments three times daily at a volume of 0.5 ml per administration for 14 consecutive days. The control group received an equivalent volume of physiological saline on an identical schedule. Locomotor activity, quantified as total travel distance (in meters), was automatically and continuously recorded over a 72 h period for a subset of mice (n = 3 per group) using the infrared beam break sensors integrated within the PRO-MRRM-8 metabolic chambers. The Meta Screen 2.3 calculated the cumulative distance traveled by each mouse for subsequent statistical analysis. The experimental process of this study was designed to control for confounding factors such as sequence and position. Specifically, the measurement sequence of administration and behavior was balanced among the groups. While the nature of the interventions prevented the implementation of researcher blinding to the group allocation during administration and outcome assessment, all data acquisition and initial analysis were performed automatically by the instrumental software to minimize potential bias.

### 2.6. Energy Metabolism Detection

Energy metabolism was evaluated using the PRO-MRRM-8 Comprehensive Laboratory Animal Monitoring System. After 14 days of the treatment period, three mice from each group were randomly placed into individual metabolic chambers. The system was operated at an air flow rate of 254 mL/min. A calibration gas mixture containing 0.492% CO_2-_ in N_2-_ (99.99% purity) was used. Key parameters including VO_2-_, carbon dioxide production (VCO_2-_), RQ, HP, food and water intake, locomotor activity (travel distance), and evaporative water loss were continuously recorded over a 72 h period under standardized conditions. Data from a representative 24 h interval were extracted for further analysis. The raw volumes of VO_2-_ and VCO_2-_ were automatically normalized by the Meta Screen 2.3 per unit of animal body weight and are expressed as mL/kg/h. The RQ was calculated as the ratio VCO_2-_/VO_2-_ and requires no further normalization. HP (in kcal/h) was directly calculated by the system based on the measured VO_2-_ and VCO_2-_ using the established equation.

### 2.7. Compositional Analysis In Vitro and Vivo

#### 2.7.1. Preparation of Samples for In Vitro Analysis

A 500 µL aliquot of the G-CQD solution was centrifuged at 13,000 rpm for 10 min at room temperature to sediment any large aggregates or insoluble particulates. The resulting supernatant was carefully collected and filtered through a 0.22 µm sterile syringe filter to ensure the complete removal of any residual nanoparticles or microbiological contamination that could compromise the UPLC-MS instrumentation. The filtrate, representing the final in vitro test sample, was directly transferred to a sterile vial for immediate UPLC-MS analysis.

#### 2.7.2. Preparation of Drug-Containing Serum

To 200 µL of drug-containing serum, 10 µL of phosphoric acid was added. The mixture was loaded onto an activated solid-phase extraction column, which was sequentially washed with 1 mL of pure water and 1 mL of 5% methanol. Elution was carried out with 1 mL of 80% methanol. The eluate was collected, dried under vacuum at 35 °C, and reconstituted in 200 µL of methanol. After vortex mixing for 30 s and sonication for 5 min, the sample was centrifuged at 13,000 rpm and 4 °C for 15 min. The supernatant was filtered through a 0.22 µm membrane prior to analysis.

#### 2.7.3. UHPLC-Q/Orbitrap/LTQ MS Analysis Conditions

Chromatographic separation was performed on a Vanquish UHPLC system (Thermo Fisher Scientific, Waltham, MA, USA) equipped with a Waters ACQUITY UPLC BEH HSS T3 column (100 mm × 2.1 mm, 1.8 µm). The mobile phase consisted of (A) 0.1% formic acid in water and (B) acetonitrile. The flow rate was 0.3 mL/min, and the column temperature was maintained at 35 °C. The gradient program was as follows: 0–1 min, 5–10% B; 1–2 min, 10–27% B; 2–5 min, 27–32% B; 5–10 min, 32–50% B; 10–12 min, 50–50% B; 12–20 min, 50–99% B.

Mass spectrometric analysis was conducted using electrospray ionization in both positive and negative ion modes. Full MS/dd-MS^2^ scanning was performed with an Orbitrap resolution of 120,000 over a mass range of m/z 70–1200. Capillary voltage was set to 3.5 kV (positive) and 2.8 kV (negative). Sheath and auxiliary gas flows were 40 and 15 arb, respectively. The ion transfer tube and vaporizer temperatures were maintained at 350 °C.

#### 2.7.4. Data Processing Methods

The identification of in vitro constituents and in vivo prototype components was performed using an untargeted metabolomics approach. In vitro compositional analysis was performed using Compound Discoverer 3.3 software. Chromatographic data from UPLC-MS analyses of in vitro G-CQDs were imported into the Compound Discoverer 3.3 platform. Using the software’s integrated natural product solution workflow, we established a custom compound library for ginseng constituents based on the Traditional Chinese Medicine Systems Pharmacology database and the relevant literature. This library was imported into Compound Discoverer 3.3. Integrating these resources, precursor and fragment ions detected in the UPLC-MS data were matched against the custom ginseng library and public phytochemical databases to efficiently annotate compounds of botanical origin.

In vivo prototype component analysis was performed based on the in vitro compositional data. Serum from the model group served as the control for the exclusion of endogenous interference. The identification of in vivo metabolites was conducted using the integrated metabolic prediction module within the Compound Discoverer 3.3 software, which served as our in silico metabolic model. This model simulates both Phase I and Phase II biotransformations based on known enzymatic reactions. The following parameters were applied: a mass accuracy threshold of 5 ppm for precursor and fragment ions; consideration of a comprehensive set of metabolic reactions, including hydrolysis, hydration, oxidation, reduction, dehydroxylation, deglycosylation, methylation, acetylation, glucuronidation, sulfation, glycine conjugation, glutamine conjugation, glutathione conjugation, and arginine conjugation; and a maximum of three sequential reaction steps. The software automatically matched the detected components against the in vitro library of G-CQD constituents and then applied this metabolic reaction network to predict and identify potential metabolites. Putative identifications were subsequently verified by examining the corresponding MS/MS fragmentation patterns to ensure structural plausibility.

### 2.8. Correlation Analysis

Correlation coefficients between energy metabolism parameters and the relative abundance of in vivo components were calculated using R software (version 3.5.1). Correlations with |*r*| > 0.7 were considered strong. Highly correlated components were visualized and identified as key bioactive constituents influencing energy metabolism.

### 2.9. Statistical Analysis

Data analysis was conducted using GraphPad Prism 8.4.0 software. Continuous data were first subjected to normality testing using the Shapiro–Wilk test, and then to homogeneity of variance testing using the Brown Forsythe test. Data that conform to a normal distribution and have equal variance are represented as mean ± standard deviation. The statistical significance is set to *p* < 0.05. Correlation analysis uses Pearson correlation coefficient (data follows normal distribution) or Spearman rank correlation coefficient (data does not follow normal distribution).

## 3. Result

### 3.1. Structural and Surface Characterization of G-CQDs

TEM was employed to examine the morphology, particle size distribution, and local crystalline structure of the as-synthesized G-CQDs. In the low-magnification TEM image, the sample shows uniformly dispersed dot-like contrast. High-resolution imaging reveals predominantly amorphous contrast with locally ordered fringes. Two ROIs extracted from Figure 1B display clearer fringes, and their 2D-FFT patterns exhibit diffuse spots/arcs whose principal radial peaks correspond to d ≈ 0.27–0.31 nm, consistent with short-range graphitic ordering in small sp^2^ domains. FTIR spectroscopy was used to identify the surface functional groups and chemical bonds present on the G-CQDs. FTIR spectroscopy shows a broad, intense band at 3386 cm^−^^1^ attributable to O–H stretching, indicating abundant surface hydroxyls that confer hydrophilicity and hydrogen-bonding sites. Weak features at 2957 and 2868 cm^−^^1^ arise from aliphatic C–H stretching, suggesting residual saturated segments on the carbon framework [18]. Strong absorption at 1708 cm^−^^1^ corresponds to C=O stretching from carboxyl/ester/amide groups, consistent with oxidative functionalization [19]. In the fingerprint region, bands at 1119 and 1047 cm^−^^1^ are assigned to skeletal C–C vibrations and C–O (alcohol/phenol/ether) stretching, respectively, and the 713 cm^−^^1^ band to C–CO–C bending/out-of-plane modes, collectively evidencing oxygen-rich surfaces (Figure 2C) [20,21,22]. UV-Vis absorption spectroscopy was employed to investigate the optical absorption properties and estimate the optical bandgap of the as-synthesized G-CQDs. Under UV excitation, ginseng-derived CQDs exhibit a single emission band peaking at 420 nm with a pronounced long-wavelength tail; the intensity decays rapidly from 500 to 700 nm, and emission above 700 nm is negligible except for a slight rise at 780–800 nm, likely due to stray light/second-order diffraction(Figure 2B). This spectral signature agrees with π–π transitions of an sp^2^ carbon core plus surface-state emission from oxygenated defects, in line with the FTIR-identified –OH, C=O, and C–O groups [23]. Zeta potential measurements were conducted to evaluate the colloidal stability and surface charge of the G-CQD dispersions. Electrophoretic measurements reveal a predominantly negative zeta-potential (weighted mean −16.4 mV; mode/median −17.6 mV; D10/D50/D90 −31.2/−17.6/−4.0 mV), with ~97.6% of particles between −30 and −5 mV, indicative of moderate colloidal stability; the negative charge arises from deprotonated oxygen functionalities and supports aqueous dispersibility, as well as downstream bioconjugation/cellular interactions (Figure 2A).

### 3.2. G-CQDs Elicit Superior Systemic Energy Metabolism Enhancement over G-AE

The metabolic cage assay quantified real-time whole-body substrate use and energy expenditure. Both interventions increased energy metabolism versus control. With G-CQDs, CO_2-_ production rose by 11.0%, HP by 14.6% (*p* < 0.05)—indicating enhanced energy expenditure, elevated basal metabolic rate, and augmented thermogenesis—RQ by 6.1% (*p* < 0.05), reflecting a greater reliance on carbohydrate-derived energy substrates, and O_2-_ consumption by 4.1%. Behavior and fluid-balance measures showed parallel elevations: evaporative water loss +18.5% (*p* < 0.05), locomotor distance +60.5% (*p* < 0.05), consistent with increased physical activity, food intake +81.6% (*p* < 0.05), and water consumption +50.8% (*p* < 0.05). Relative to G-AE, G-CQDs further improved multiple endpoints, including CO_2-_ (+3.2%, *p* < 0.05), HP (+6.7%), RQ (+2.8%), evaporative water loss (+4.6%, *p* < 0.05), locomotor distance (+22.9%, *p* < 0.05), food intake (+27.9%), and water consumption (+10.0%, *p* < 0.05); O_2-_ consumption was similar (−0.9%). Together, these effects indicate that CQD nano-delivery elicits a higher whole-body energy turnover—via greater thermogenesis, a carbohydrate-biased fuel mix, and behavioral activation—than conventional ginseng extract (Figure 3).

### 3.3. Comprehensive Mass Spectrometry Profiling Identifies 110 Chemical Constituents in G-CQDs’ Nanocomplex

A comprehensive chemical characterization of G-CQDs was achieved using UHPLC-Q-Orbitrap/LTQ MS, yielding high-resolution MS and MS/MS data. In positive ion mode, quasi-molecular ions were observed as [M+H]^⁺^, [M+Na]^⁺^, [M+H+H_2-_O]^⁺^, and [M-e]^⁺^ adducts; in negative ion mode, [M-H]^−^, [M-H+H_2-_O]^−^, [M+HCOO]^−^, and [M+e]^−^ species were detected. The base peak intensity chromatogram is shown in Figure 4. Within a 20 min chromatographic run, 110 distinct chemical constituents were identified (Appendix A).

The mass spectrometric characterization of ginsenoside Rg5 exemplifies the structural elucidation workflow for G-CQDs, performed using Compound Discoverer 3.3 (CD3.3) software. Following automatic noise reduction, peak alignment, and feature extraction in negative ion mode, a precursor ion at m/z 785.4777 (RT 15.60 min) was detected, corresponding to the molecular formula C_42-_H_70_O_12-_. Higher-energy collisional dissociation of this precursor yielded diagnostic fragment ions at m/z 603.4257, 441.3746, 221.0621, 161.0453, 113.0241, 101.0240, 85.0292, and 71.0139. Fragmentation analysis indicated neutral losses of hexose units (C_6_H_10_O_5_), generating ions at m/z 603.4257 and 441.3746, while successive cleavages of C_34_H_5__6_O_5_, C_2-_H_4_O_2-_, CH_3_O_2-_, CH, O, and C produced ions at m/z 221.0621 through 71.0139. This fragmentation pattern demonstrated complete concordance with the reference MS/MS spectrum of an authentic ginsenoside Rg5 standard within the CD3.3 database. The compound was, thus, unequivocally identified as ginsenoside Rg5, with its proposed fragmentation pathway illustrated in Figure 5.

### 3.4. In Vivo Tracing Reveals 35 Prototype Components of G-CQDs

A strategy combining UHPLC-Q/Orbitrap/LTQ MS with multimodal data processing was established to systematically identify prototype components of G-CQDs circulating in the blood. First, in data-dependent acquisition mode, blank serum was designated for the exclusion list, and drug-containing serum was designated for the inclusion list. Characteristic peaks were comprehensively captured by repeated injections of drug-containing serum samples, effectively excluding interference from the blank matrix. Based on the in vitro chemical composition data, the molecular ion peaks of the identified constituents were extracted, and potential in vivo prototype components were screened by matching retention times (RT). Chromatograms of serum samples from G-CQDs and G-AE intervention mice, collected in positive and negative ion modes, are shown in Appendix A. Thirty-five prototype components were identified in serum following treatment with G-CQDs (Appendix A).

The identification of quinic acid by mass spectrometry was used to demonstrate the resolution process for organic acids, which are the main chemical components of G-CQDs, based on CD3.3 software. In negative ion mode, the molecular ion peak at m/z 191.0566 was extracted at RT = 0.84 min following automatic noise reduction, peak matching, and extraction by the CD 3.3 software. Its molecular formula was presumed to be C_7_H_12-_O_6_. The main fragment ions generated by this parent ion under high-energy collisional dissociation were m/z 173.0451, 155.0346, 137.0241, 93.0346, 87.0084, 71.0136, 59.0136, and 44.9979. Fragmentation analysis revealed that ion fragments m/z 173.0451, 155.0346, 137.0241, and 93.0346 were obtained by sequential removal of H_2_O, H_2_O, H_2_O, and CO_2_ by the parent ion, ion fragment m/z 87.0084 was obtained by the parent ion by removing C_4_H_8_O_3_, ion fragments m/z 71.0136 and m/z 59.0136 were obtained from fragment m/z 87.0084 by removing O and CO, respectively, and ion fragment m/z 59.0136 was further removed from C to obtain fragment ion m/z 44.9979. The above fragment ions exhibited a high degree of consistency with the secondary fragmentation information of quinic acid standards in the CD3.3 software database. Consequently, the compound was identified as quinic acid, and its potential cleavage pathway is illustrated in Figure 6.

### 3.5. Metabolic Mapping Identifies 29 Phase I and II Metabolites of G-CQDs

The Compound Discoverer 3.3 software platform facilitates not only the accurate identification of drug components, but also the analysis of potential metabolic pathways and the characterization of phase I and phase II metabolites generated from prototype compounds. A total of 29 drug metabolites were characterized in the serum of G-CQD-intervened mice, of which phase I metabolism mainly includes deglycosylation, dehydration, reduction, oxidation, hydration, and desaturation pathways, while phase II metabolism mainly includes methylation, acetylation, sulphation, glutamine-binding, glutathione-binding, glucuronide-binding, and arginine-binding pathways. The specific metabolite information is shown in Appendix A. The identification process is illustrated using metabolite M2, a derivative of ginsenoside Re.

Metabolite M2 exhibited an RT of 8.52 min and a molecular ion peak at m/z 561.2961[M-2H]^2−^. Key fragment ions were observed at m/z 1077.5726, 945.5297, 783.4803, 621.4316, and 265.6219, of which m/z 945.5297 was the typical parent ion peak of ginsenoside Re under the negative ion model, and m/z 783.4803, m/z 621.4316, and m/z 945.5297 differed from m/z 945.5297 by 162 Da and 324 Da, which are secondary fragments obtained by its successive deletion of the two glycosidic bonds C_6_H_10_O_5_. Furthermore, the fragment at m/z 1077.5726 was consistent with a loss of -CHO_2_ from the precursor, and the fragment at m/z 265.2619 was consistent with a loss of C_45_H_70_O_16_. These fragmentation patterns showed high agreement with the reference MS/MS data for ginsenoside Re in the database. The possible cleavage pathway of metabolite M2 is shown in Figure 7. Based on the mass spectrometry cleavage pathway combined with the secondary mass spectra of the metabolites, 11 metabolites of ginsenoside Re were identified, and the possible metabolic pathways of ginsenoside Re in vivo were presumed to be deglycosylation, dehydration, reduction, oxidation, hydration, desaturation, glucuronide conjugation, glycine conjugation, methylation, acetylation, and arginine conjugation. The specific metabolic pathways are shown in Figure 8.

### 3.6. G-CQDs Markedly Enhance Systemic Exposure of Bioactive Components In Vivo

Bivariate correlation analysis on the Metware Cloud platform assessed associations between serum components and energy metabolism indices, revealing eight core bioactive components strongly linked to enhanced energy metabolism parameters (Pearson |R| > 0.7, *p* < 0.05). This analysis revealed potential bioactive components associated with G-CQDs’ pro-metabolic effects, visualized in Figure 9. Eight core bioactive components were identified. The systemic exposure of all eight components was higher in the G-CQDs group than in the G-AE group, with the content of G-CQDs increasing relative to G-AE as follows: Protopanaxadiol+C_2_H_2_O (+89.2%), Ginsenoside Rb1+H_2_O+C_6_H_8_O_6_ (+105.0%), Linolenic acid+H_2_+C_10_H_17_N_3_O_6_S (+330.8%), Ginsenoside Rh1+O+O+CH_2_ (+42.1%), Ginsenoside Re (+115.1%), Ginsenoside Rg1 (+8.0%), Cinnamic acid+H_2_O+CH_2_ (+27.5%), and Cinnamic acid+O+SO_3_ (+108.8%). Specifically, the levels of ginsenoside Re, ginsenoside Rg1, and specific modified forms of cinnamic acid, ginsenoside Rb1, and linolenic acid were significantly higher in the G-CQDs group (*p* < 0.05). These findings demonstrate that G-CQDs enhance the bioavailability of ginseng saponins and facilitate the formation of unique phase II conjugates, providing a molecular basis for their superior modulation of energy metabolism compared to traditional ginseng preparations.

## 4. Discussion

This study successfully developed a novel G-CQD delivery system by transforming traditional ginseng preparations into nanoformulations. Using ultrahigh-speed nitrogen flow pulverization and far-infrared pulsed carbonization, poorly soluble bioactive components like ginsenosides were encapsulated within a carbon nano-skeleton. This unique preparation method likely contributed to the formation of a nanostructure with abundant surface functional groups, which is crucial for its subsequent performance [24]. This approach effectively overcomes the bioavailability limitations inherent to prototype saponins, primarily by enhancing their aqueous dispersibility and potentially improving intestinal absorption, a common challenge for many herbal bioactive compounds [25]. Animal experiments confirmed that G-CQDs significantly enhance energy metabolism, with superior promotion of RQ and HP compared to G-AE. Integrated high-resolution mass spectrometry and multivariate correlation analysis enabled the identification of 110 in vitro components, 35 blood-circulating prototype components, and 29 metabolites. Notably, eight core bioactive components associated with energy metabolism enhancement were identified, all exhibiting higher systemic exposure in the G-CQDs group than the G-AE group. These findings highlight the potential of herbal nano-transformation at the component–metabolome interface.

Unlike conventional CQDs, herb-derived CQDs retain the bioactive constituents and medicinal properties of their source materials. The application of CQDs in traditional medicine has revealed novel pharmacodynamic effects. For example, Selaginella-derived CQDs enhance coagulation and fibrinogen activation [26], while Lonicera japonica-derived CQDs alleviate inflammation [27]. Artemisia argyi-derived CQDs exhibit antibacterial activity [28], and coriander leaf-derived CQDs demonstrate antioxidant properties [29]. Previous reports indicate that G-CQDs can target the meningeal system, scavenge ROS, chelate iron, improve hemodynamics, and confer neuroprotection and antitumor effect [28,30,31,32]. Using continuous energy phenotyping, we demonstrated that G-CQDs significantly enhance metabolic activation, indicating novel pharmacological potential for ginseng nanoformulations. The G-CQD platform established in this study provides a versatile and scalable strategy that could be extended to enhance the bioavailability and efficacy of other challenging phytochemicals beyond ginseng. This approach holds significant promise for a wide range of poorly bioavailable natural products, such as curcuminoids (from turmeric), flavonoids (e.g., quercetin), and other hydrophobic saponins, thereby modernizing the delivery of natural product-based therapeutics. Furthermore, this study exemplifies a sustainable and modernized processing technique for TCM. By transforming the entire herb into a bioavailable nano-complex using aqueous-based synthesis, it enhances resource efficiency and minimizes waste. This approach modernizes TCM by creating a well-defined, quality-controllable nano-formulation, effectively addressing the long-standing challenges of low bioavailability and batch-to-batch variability.

Our G-CQDs are distinct from existing nanomaterials in composition and function. Unlike extracted ginseng nanoparticles, they are carbonized, smaller constructs with embedded bioactives. In contrast to synthetic carbon carriers serving as inert drug vehicles, G-CQDs are inherently bioactive and self-therapeutic [33]. Their native surface groups further eliminate post-synthetic modification needs, enabling their primary role as bioavailability-enhancing carriers with demonstrated efficacy beyond conventional CQD applications [34]. A defining attribute of carbon quantum dots is their rich surface functionality (–OH, –COOH, –NH_2-_), which governs water dispersibility, colloidal stability, interfacial interactions, biocompatibility, and load/adsorption behavior. Our FTIR/ζ-potential data (oxygen-rich surfaces; negative ζ) are consistent with this paradigm and mechanistically explain (i) stable dispersions during oral dosing and (ii) noncovalent interactions (H-bonding/π–π) with ginseng small molecules that can retain/solubilize poorly soluble saponins in the GI milieu. Contemporary reviews converge on this structure–property link for CQDs in drug delivery [35]. Beyond simple solubilization, biological identity acquired via protein/mucin corona in the gastrointestinal tract can modulate mucodiffusion, epithelial contact time, and uptake. Studies on carbon dots show that surface charge and functional groups reshape corona composition and downstream biological responses, while mucin corona formation in simulated GI fluids alters mobility and interactions—factors likely relevant to our higher systemic exposure [36]. These literature observations support the view that CQDs’ surface chemistry is a tunable handle to optimize oral fate [37].

Energy metabolism underpins all biological processes, regulating the anabolism and catabolism of carbohydrates, lipids, and proteins [38]. Disruptions in energy metabolism have become a common pathological basis for metabolic diseases including obesity, type 2 diabetes, and non-alcoholic fatty liver disease [11,12,13]. Natural active components, particularly those from ginseng, have attracted attention for their ability to modulate energy metabolism. Ginseng, a revered herb for “tonifying primordial Qi” (DaBuYuanQi), regulates energy metabolism by promoting glucose-derived energy production and inhibiting fatty acid metabolism [39]. This study identifies key active components within G-CQDs that are responsible for regulating energy metabolism. These eight core bioactive clusters are predominantly ginsenoside prototypes (e.g., ginsenoside Re, Rg1) and their phase II metabolites (e.g., sulfated, glucuronidated, and glutathione-conjugated derivatives), alongside modified forms of organic acids such as cinnamic acid. This composition underscores the central role of both native saponins and their biotransformed products in mediating the energy metabolism-enhancing effects of G-CQDs. Ginsenoside Rb1 improves insulin resistance and hepatic steatosis through the activation of the STAT3 pathway, promoting glycolytic enzyme GLUT2 and suppressing gluconeogenic enzyme PEPCK [40,41]. Ginsenoside Rg1 enhances mitochondrial oxidative phosphorylation (OXPHOS) efficiency and ATP production, while cinnamic acid boosts OXPHOS function and respiratory chain expression [42]. Additionally, ginsenoside Re normalizes hypothalamic–pituitary–adrenal axis function and improves hepatic gluconeogenesis and muscle glucose uptake. Linolenic acid maintains mitochondrial cardiolipin homeostasis, thereby supporting mitochondrial function and energy metabolism [42,43]. Collectively, G-CQDs regulate energy metabolism through a multi-target, multi-level mechanism: enhancing glucose metabolism, preserving mitochondrial function, improving OXPHOS efficiency, and activating respiratory chains. Future studies will employ mass spectrometry imaging to map the distribution of key bioactive components in target organs, utilizing multi-omics approaches to further elucidate their mechanistic role in energy metabolism. This research could offer new therapeutic strategies for metabolic diseases such as obesity, diabetes, and non-alcoholic fatty liver disease.

The complex chemical composition of TCM poses challenges for comprehensive characterization [44]. TCM contains a diverse array of compounds, yet only those that are absorbed into circulation exert pharmacological effects. The data analysis method in this study facilitates rapid chemical analysis of in vivo components in Chinese herbal preparations [45]. Using UHPLC-Q/Orbitrap/LTQ MS technology, we identified 110 chemical components in G-CQDs in just 20 min. Through a combination of blank serum exclusion lists and dosed serum inclusion lists to eliminate endogenous interference, 35 prototype components were identified in vivo. Phase I/II metabolic pathways were simulated using CD 3.3, leading to the identification of 29 metabolites in G-CQDs. This rapid identification framework provides a new paradigm for complex TCM system research.

While this study provides compelling proof-of-concept for G-CQDs as a bioavailability-enhancing nano-carrier, several aspects remain unclear and warrant further investigation, which also represents the direction of our future research. First, beyond demonstrating excellent aqueous stability, our physicochemical characterization indicated a high negative zeta potential and a narrow, monodisperse size distribution—features that support colloidal stability and help maintain formulation integrity during storage and oral administration. Notably, the current stability assessment was performed only in aqueous dispersion; future work will implement comprehensive, physiologically relevant stress-tests (simulated gastric/intestinal fluids, as well as stability across different pH levels and buffer systems) to better predict in vivo behavior and facilitate clinical translation. Second, the precise mechanisms underlying intestinal absorption and in vivo trafficking of G-CQDs are yet to be elucidated; forthcoming studies will use mass spectrometry imaging to map biodistribution to key metabolic organs (e.g., liver, muscle, brown adipose tissue) and, in parallel, leverage the intrinsic fluorescence of G-CQDs for real-time tracking of biodistribution and target engagement in vivo. Third, although the primary focus here was pharmacology and bioavailability (and photoluminescence quantum yield quantification was, therefore, out of scope), the inherent optical properties of G-CQDs were not characterized. Future work will determine the absolute photoluminescence quantum yield and undertake detailed spectroscopic analyses—particularly of excitation-dependent emission—to clarify emission mechanisms and to evaluate utility not only as fluorescent probes for bioimaging and in vivo tracing, but also for biosensing of specific biological molecules [46]. Finally, the long-term biosafety and potential toxicity of G-CQDs upon repeated administration require systematic evaluation. A comprehensive toxicological profile, including investigations into potential genotoxicity and the extent of organ accumulation (particularly in the liver and kidneys as primary clearance organs), will be essential prior to any clinical translation. Beyond these, the ROS-regulating dual functionality (generation and scavenging) reported for engineered CQDs suggests that G-CQDs may similarly modulate cellular redox states, a hypothesis that could help explain their observed bioactivity and merits focused investigation [47,48]. While this study provides a robust proof-of-concept, the translational pathway for G-CQDs necessitates addressing scale-up production and safety. The synthesis method, relying on ultrahigh-speed pulverization and pulsed carbonization, is inherently suitable for process control and scaling. Future work will focus on optimizing this process under Good Manufacturing Practice guidelines to ensure batch-to-batch consistency.

## 5. Conclusions

In summary, this study successfully developed a novel G-CQD nano-delivery system through an integrated approach of ultrahigh-speed nitrogen jet pulverization and far-infrared pulse-assisted hydrothermal carbonization. Crucially, our in vivo energy metabolism phenotyping provided direct physiological evidence that this nano-formulation profoundly outperforms conventional G-AE. The G-CQDs elicited a comprehensive enhancement of whole-body energy turnover in mice, characterized by a significant elevation in HP (+14.6% vs. control), a marked shift towards carbohydrate oxidation (RQ +6.1%), and a substantial boost in spontaneous locomotor activity (+60.5%). The superior efficacy of G-CQDs is unequivocally attributed to their ability to dramatically enhance the systemic exposure of ginseng’s bioactive components. Our integrated chemical profiling and correlation analysis identified eight core bioactive clusters responsible for the metabolic benefits. Notably, the nano-formulation led to a striking increase in the systemic levels of these clusters—exemplified by ginsenoside Re (+115.1%), a unique cinnamic acid–sulfate conjugate (+108.8%), and a linolenic acid–glutathione adduct (+330.8%)—compared to the G-AE group. Therefore, this work establishes a direct cause-and-effect relationship: the CQD-enabled enhancement in bioavailability directly translates to a more potent in vivo pharmacological response. Future studies should generalize the G-CQD platform to additional low-bioavailability phytotherapeutics while elucidating the mechanisms underlying its enhanced exposure—encompassing cellular uptake routes, intracellular fate, and in vivo trafficking. In parallel, scalable, quality-controlled manufacturing with long-term safety and PK/biodistribution evaluation will be essential for translation, and the therapeutic utility of G-CQDs in rational combination regimens for metabolic disease warrants rigorous assessment.

## 6. Limitations

Although the C57BL/6 mice used in this study are a common model in research, they differ from humans, which may limit the direct extrapolation of our findings to human clinical scenarios. Additionally, we did not evaluate the potential effects of age and sex on the results. The sample size of this study was determined based on preliminary experiments and literature reports. Despite our efforts, the sample size remains relatively small, which may result in insufficient statistical power to detect subtle yet biologically significant effects. Nevertheless, we believe that the core conclusions of this study are reliable. Future research will focus on validating these findings in more species or animals of different sexes, increasing the sample size to further verify and extend the discoveries of this study.

## Figures and Tables

**Figure 1 pharmaceutics-17-01485-f001:**
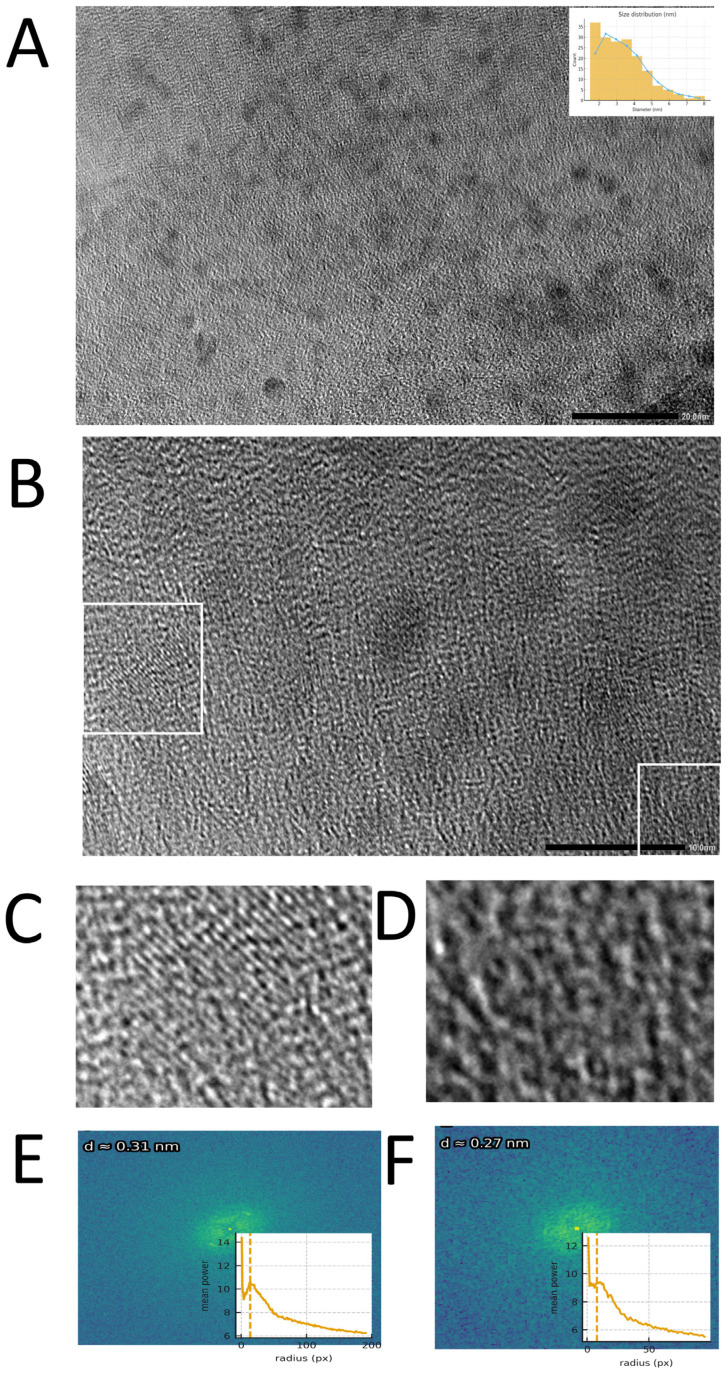
TEMHRTEM characterization of G-CQDs. (**A**) Low-magnification TEM overview (scale bar: 20 nm) with size histogram inset (percentage, diameter in nm), showing a distribution mainly between 1 and 2 nm. (**B**) HRTEM field (scale bar: 10 nm) with two ROIs highlighted. (**C**,**D**) Magnified HRTEM of the ROIs extracted from (**B**). (**E**,**F**) 2D-FFT (log amplitude) of (**C**,**D**), whose principal radial peaks give d ≈ 0.27–0.31 nm, indicative of short-range graphitic sp^2^ ordering.

**Figure 2 pharmaceutics-17-01485-f002:**
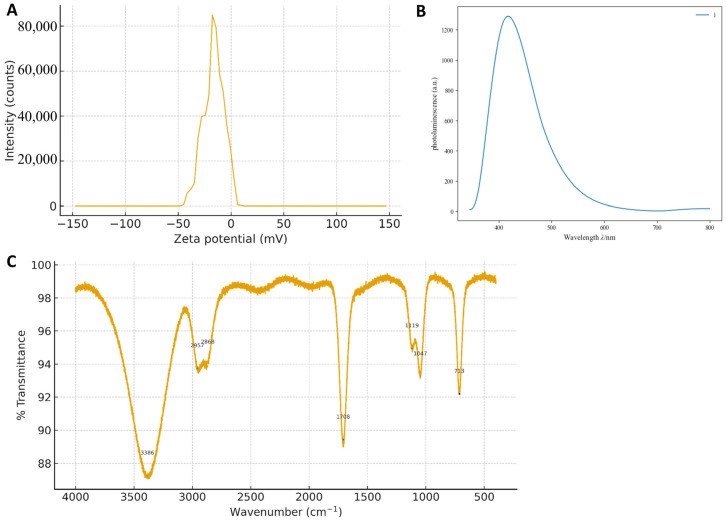
Physicochemical characterization of G-CQDS. (**A**) Zeta-Potential: predominantly negative (mean −16.4 mV; mode/median −17.6 mV; D10/D50/D90 = −31.2/−17.6/−4.0 mV; ~97.6% within −30 to −5 mV), indicating moderate colloidal stability. (**B**) UV–Vis absorption spectrum: Y-axis: Absorbance (a.u.); X-axis: Wavelength (nm). Maximum absorption at λ_max = 420 nm. (**C**) FTIR: 3386 cm^−^^1^ (O–H), 2957/2868 cm^−^^1^ (aliphatic C–H), 1708 cm^−^^1^ (C=O), 1119/1047 cm^−^^1^ (C–C/C–O), and 713 cm^−^^1^ (C–CO–C), evidencing oxygen-rich surfaces.

**Figure 3 pharmaceutics-17-01485-f003:**
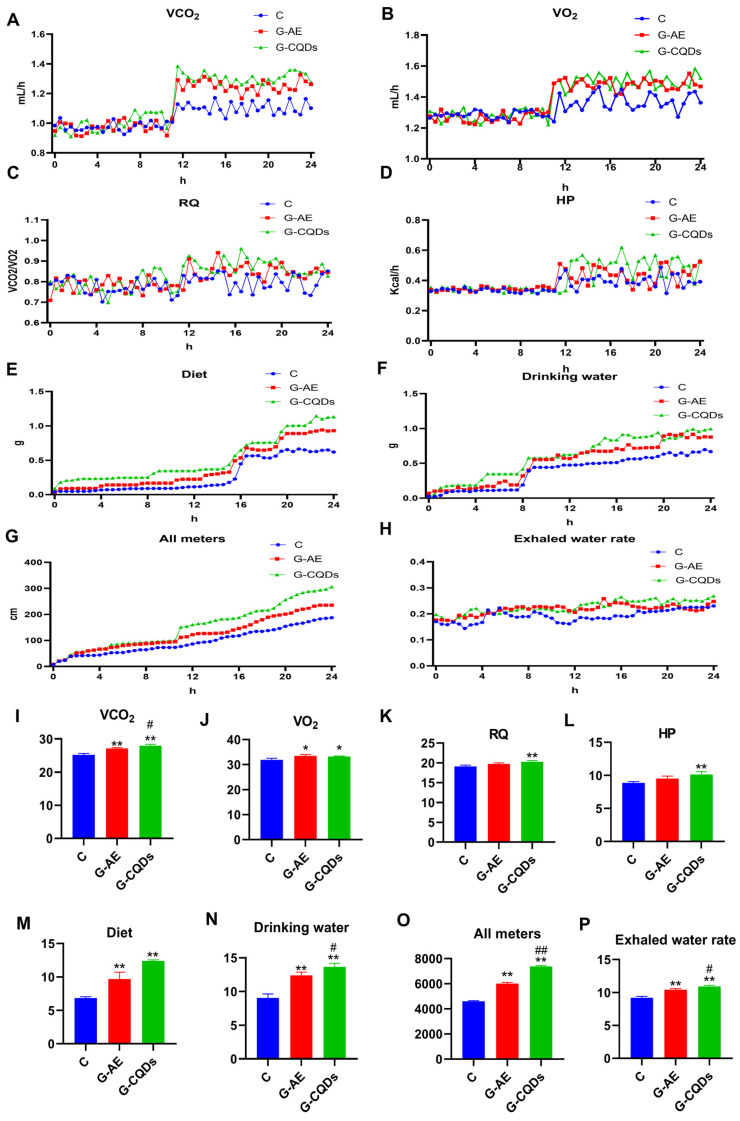
Effects of G-CQDs on murine energy metabolism and behavioral parameters. (**A**–**H**) Temporal profiles: (**A**) VO_2-_ rate; (**B**) VCO_2-_ rate; (**C**) RQ; (**D**) HP; (**E**) food intake; (**F**) water consumption; (**G**) locomotor distance; and (**H**) evaporative water loss. (**I**–**P**) Area under the curve (AUC) analysis: (**I**) VO_2-_ rate AUC; (**J**) VCO_2-_ rate AUC; (**K**) RQ AUC; (**L**) HP AUC; (**M**) food intake AUC; (**N**) water consumption AUC; (**O**) locomotor distance AUC; and (**P**) evaporative water loss AUC. Data represent mean ± SD (n = 3). Statistical significance versus control group: * *p* < 0.05, ** *p* < 0.01; versus G-AE group: # *p* < 0.05, ## *p* < 0.01.

**Figure 4 pharmaceutics-17-01485-f004:**
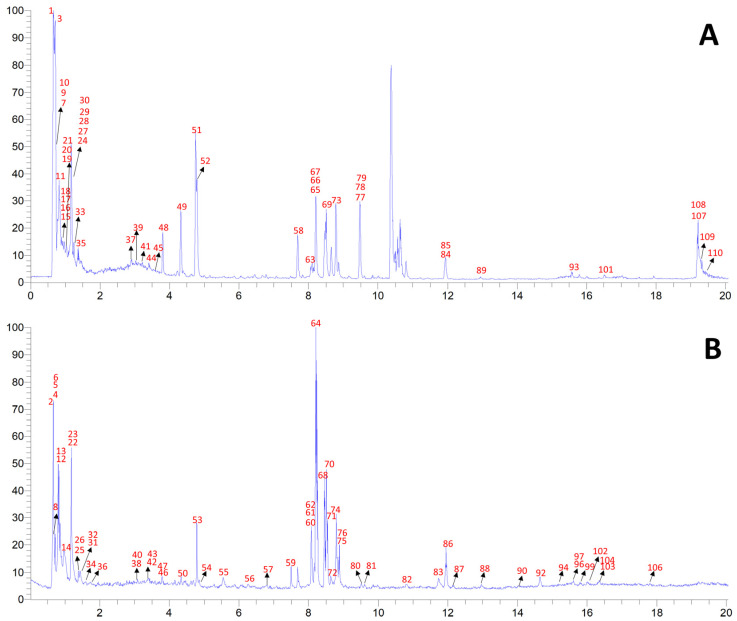
Chromatogram of in vitro chemical composition analysis of G-CQDs in positive and negative ion mode. ((**A**): positive ion mode (**B**): negative ion mode).

**Figure 5 pharmaceutics-17-01485-f005:**
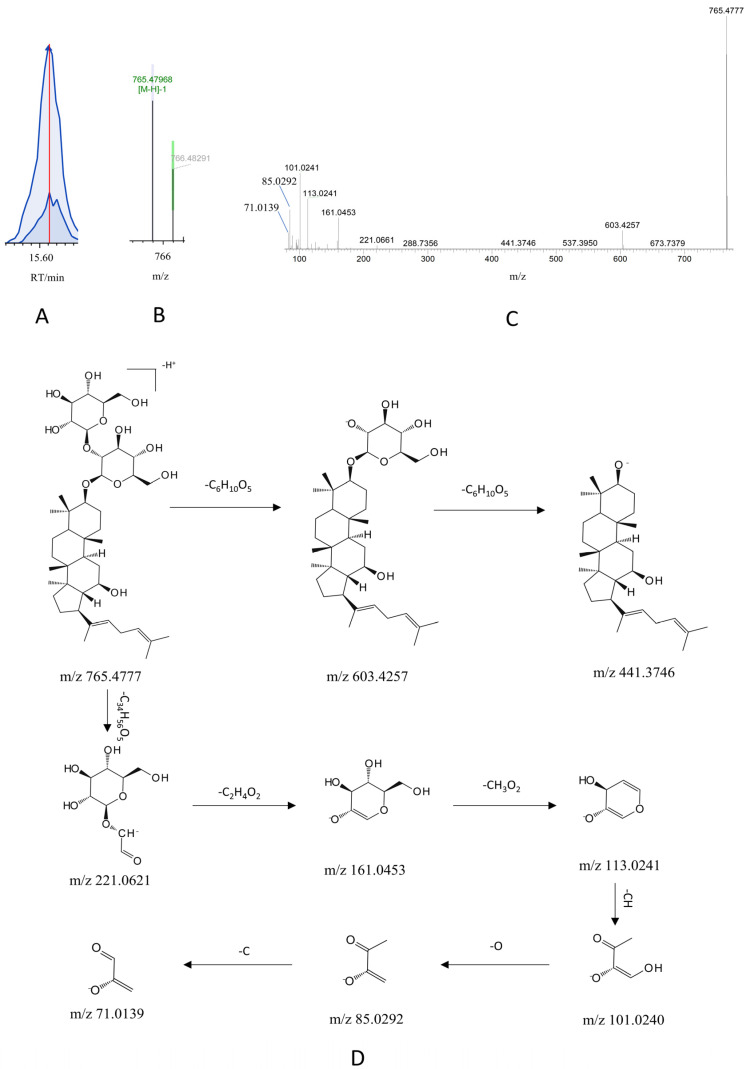
Identification and cleavage pathway analysis of ginsenoside Rg5 based on CD 3.3 software platform. (**A**) Chromatographic peak extraction of ginsenoside Rg5 in negative ion mode. (**B**) Primary mass spectrometry of ginsenoside Rg5 in negative ion mode. (**C**) Secondary mass spectrometry of ginsenoside Rg5 in negative ion mode. (**D**) Proposed fragmentation pathway of ginsenoside Rg5.

**Figure 6 pharmaceutics-17-01485-f006:**
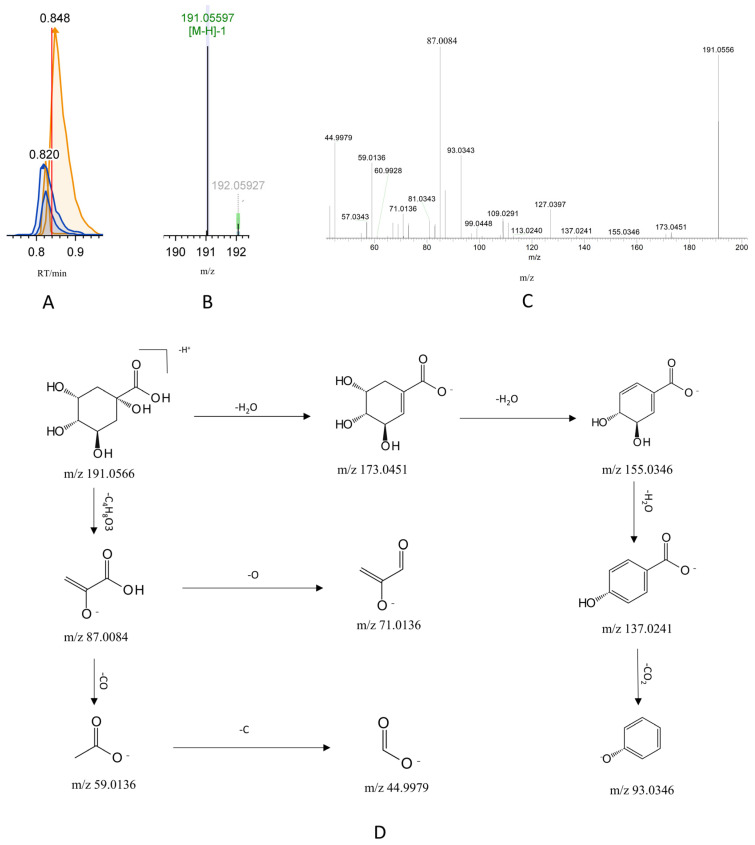
Identification and cleavage pathway analysis of quinic acid based on CD3.3 software platform. (**A**) Chromatographic peak extraction of quinic acid in negative ion mode. (**B**) Primary mass spectrometry of quinic acid in negative ion mode. (**C**) Secondary mass spectrometry of quinic acid in negative ion mode. (**D**) Proposed fragmentation pathway of quinic acid.

**Figure 7 pharmaceutics-17-01485-f007:**
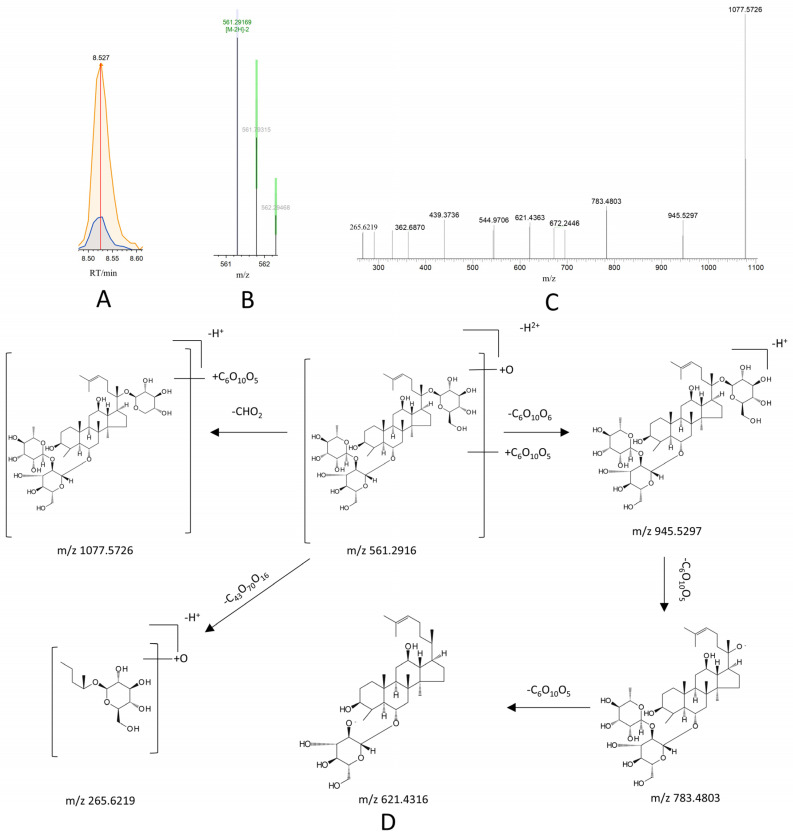
Identification and cleavage pathway analysis of metabolite M2 based on CD3.3 software platform. (**A**) Chromatographic peak extraction of metabolite M2 in negative ion mode, (**B**) primary mass spectrometry of metabolite M2 in negative ion mode, (**C**) secondary mass spectrometry of metabolite M2 in negative ion mode, and (**D**) proposed fragmentation pathway of metabolite M2.

**Figure 8 pharmaceutics-17-01485-f008:**
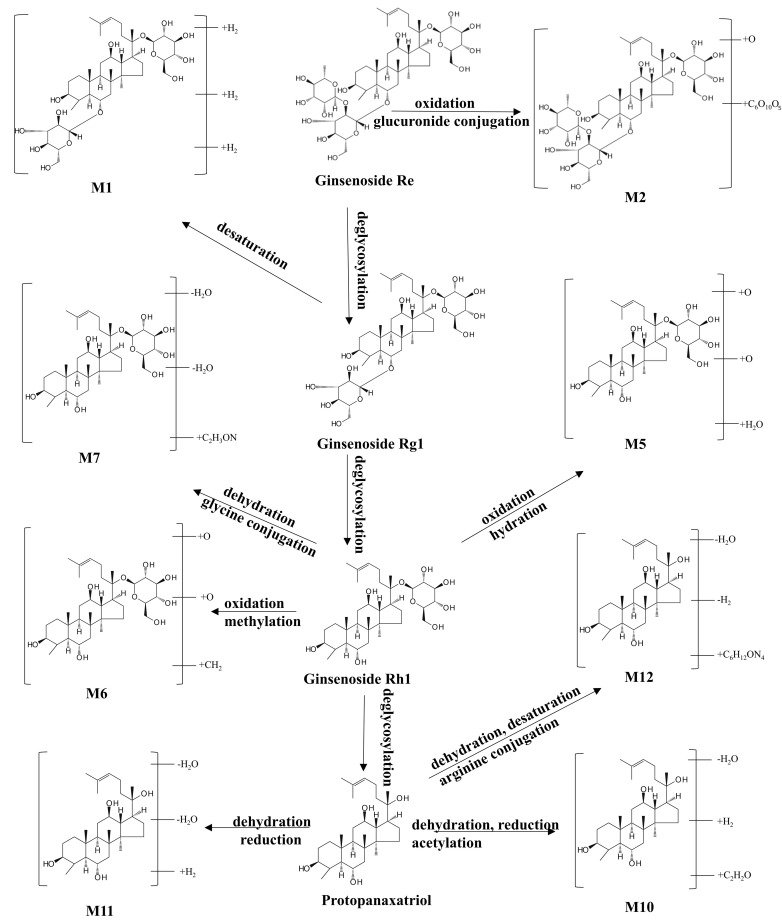
Possible metabolic pathways of ginsenoside Re in vivo.

**Figure 9 pharmaceutics-17-01485-f009:**
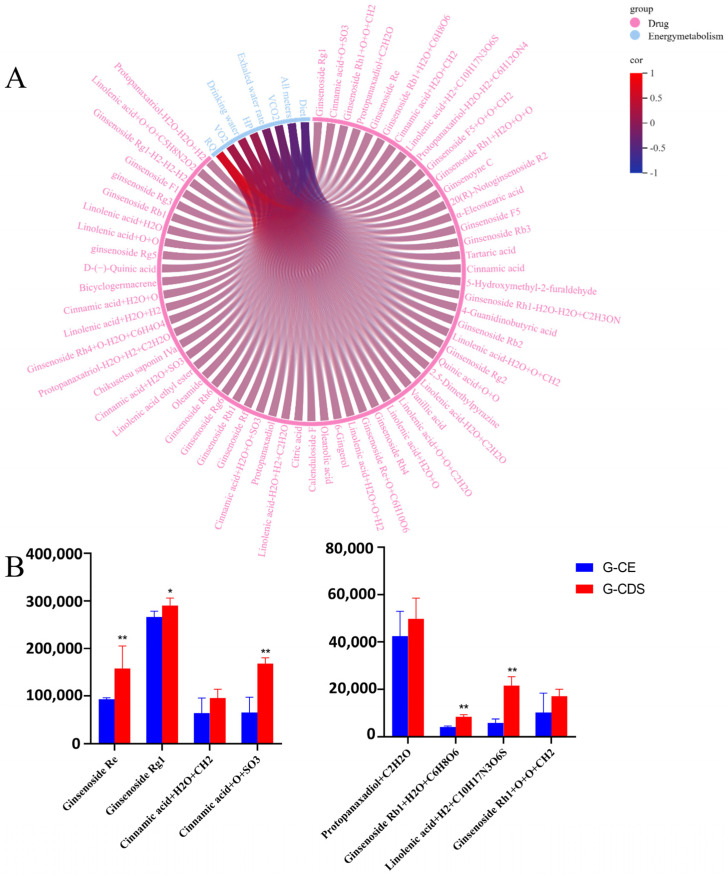
Enhanced systemic exposure of ginseng bioactives via CQDs technology. (**A**) Correlation matrix visualizing significant associations between blood-circulating components and energy metabolism enhancement in G-CQDs-treated mice. (**B**) Relative abundance of bioactive components in systemic circulation following G-CQDs versus G-AE administration. (x ± SD, n = 6, compared with G-AE, * *p* < 0.05, ** *p* < 0.01).

## Data Availability

The original contributions presented in this study are included in the article/Appendix A. Further inquiries can be directed to the corresponding authors.

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
