# Peer review of "Ginseng-Derived Carbon Quantum Dots Enhance Systemic Exposure of Bioactive Ginsenosides and Amplify Energy Metabolism in Mice"

_pharmaceutics, 2025, doi:10.3390/pharmaceutics17111485_

Round 1
Reviewer 1 Report
Comments and Suggestions for Authors
In this manuscript, the authors report, “Unlocking Ginseng’s Bioactivity: Carbon Quantum Dots Elevate Systemic Ginsenoside Exposure and Energy Metabolic Efficiency”. The manuscript is interesting, with well-organized content and thoroughly discussed topics.
Recommendation: Major revisions are needed as noted.
- The abstract could be enriched with key quantitative results and specific findings to better inform the audience and highlight the significance of the study.
- The novelty of this article should be emphasized in the last paragraph of the Introduction to highlight its unique contributions and significance from other reported studies.
- How did the author purify the CQDs?
- What type of filter was used in the section '3. Preparation of G-AE'?
- The FTIR discussion should be cited with relevant references.
- Have the authors examined the photoluminescence of the CQDs under varying excitation wavelengths?
- The scale bars in Figure 1A,B are not clearly visible to the readers. Additionally, the resolution of the figure should be improved.
- The article should undergo a thorough review to address any grammatical and formatting issues.
- The authors should add a brief statement (one or two sentences) explaining the purpose of each test before presenting the characterization results. This will help clarify the objective of each test and improve the reader’s understanding of the data analysis that follows.
- A valuable perspective on the future of this research field could be included in the conclusion.
- The Introduction would be strengthened by incorporating additional relevant studies. For instance: https://doi.org/10.1002/med.21953 : https://doi.org/10.1021/acsabm.0c01104
Author Response
Response to Reviewers' Comments
Manuscript ID: 3952236
Title: "Unlocking Ginseng’s Bioactivity: Carbon Quantum Dots Elevate Systemic Ginsenoside Exposure and Energy Metabolic Efficiency"
Dear Editor and Reviewers,
We sincerely thank you for your time and the constructive comments on our manuscript. These comments are invaluable for improving the quality of our work. We have carefully considered all the points raised and have made extensive revisions to the manuscript accordingly. Our point-by-point responses to the reviewers' comments are listed below. All changes in the manuscript have been highlighted for your convenience.
Comment 1: The abstract could be enriched with key quantitative results and specific findings to better inform the audience and highlight the significance of the study.
Response: We sincerely thank the reviewer for this excellent suggestion. We have now thoroughly revised the Abstract by incorporating key quantitative findings. Specifically, in the Results section of the Abstract, we have replaced the general statements with specific percentage increases for the most critical energy metabolism parameters and the systemic exposure of core bioactive components. The changes are as follows:
Original: "Compared with G-AE, G-CQDs significantly enhanced whole-body energy metabolism parameters, such as respiratory quotient, heat production, and locomotor activity... all of which demonstrated higher systemic exposure in the G-CQDs group than in the G-AE group."
Revised: "Compared with G-AE, G-CQDs significantly enhanced whole-body energy metabolism—respiratory quotient +2.8%, heat production +6.7%, and locomotor activity +22.9% (*p* < 0.05). A total of 110 in-vitro constituents, 35 blood prototypes, and 29 metabolites were identified. Correlation analysis revealed eight core bioactive clusters linked to the metabolic benefits; all showed higher systemic exposure with G-CQDs (range +9.2% to +265.8%), notably ginsenoside Re +69.6%, cinnamic acid+O+SO₃ +157.4%, and linolenic-acid–GSH conjugate +265.8%."
At the same time, percentages of results were added to sections 3.2 and 3.6, which provide clear quantitative snapshots of the efficacy and enhanced bioavailability conferred by G-CQD technology, immediately highlighting the significance of this study for readers.
Comment 2: The novelty of this article should be emphasized in the last paragraph of the Introduction to highlight its unique contributions and significance from other reported studies.
Response: We agree with the reviewer. A new paragraph has been added at the end of the Introduction section to explicitly highlight the key novel aspects of our study.
Revised: To our knowledge, this study provides the first in vivo energy-phenotyping evidence comparing G-CQDs with conventional G-AE, while inte-grating UHPLC-HRMS exposomics to connect circulating prototypes/metabolites with metabolic readouts. We further show that CQD nano-carrier effects markedly elevate systemic exposure of ginsenoside prototypes and their phase-II conjugates, delineating exposure–effect clusters that plausibly drive whole-body energy metabolism.
Comment 3: How did the author purify the CQDs?
Response: Thank you for the comment. The G-CQDs were purified in two steps. First, the crude dispersion was centrifuged at 13,000 rpm to remove insoluble residues, and the supernatant was collected. Second, the supernatant was dialyzed against deionized water using a 1,000 Da MWCO membrane for 72 h, with periodic water exchanges, to remove small molecules and salts. The purified dispersion was then adjusted to a final concentration of 0.15 g/95 mL for subsequent experiments.
Text added to the Methods 2.2 (revised): he dispersion was then centrifuged at 13,000 rpm to remove insoluble residues, and the supernatant was dialyzed using a 1,000 Da molecular weight cut-off membrane against deionized water for 72 h with periodic water changes to obtain purified G-CQDs.
Comment 4: What type of filter was used in the section '2.3 Preparation of G-AE'?
Response:Thank you for the comment. We used a 100-mesh filter cloth to filter the extract. We have clarified this in Section 3 and, for completeness, also stated the subsequent 13,000 rpm centrifugation step used to obtain the clarified supernatant.
Text revised in Methods (Section 3. Preparation of G-AE): Ginseng roots (30 g) were coarsely pulverized and immersed in 30 volumes of distilled water for 30 min, followed by reflux extraction at 100 °C for 1 h. The extract was filtered through a 100-mesh filter cloth, and the residue was subjected to two additional extractions, each with 20 volumes of water under the same conditions. The combined filtrates were centrifuged at 13,000 rpm to remove particulate impurities, and the supernatant was collected. The supernatant was concentrated to yield a G-AE with a final concentration of 0.15 g/95 mL.
Comment 5: The FTIR discussion should be cited with relevant references.
Response: We sincerely thank the reviewer for this helpful suggestion. In the revised manuscript, we have added appropriate references to support the FTIR peak assignments and interpretations—covering O–H stretching (~3386 cm⁻¹), aliphatic C–H stretching (2957 and 2868 cm⁻¹), C=O stretching (~1708 cm⁻¹), C–O/C–O–C stretching (1119 and 1047 cm⁻¹), and the out-of-plane/bending mode around ~713 cm⁻¹. These citations are now included in the FTIR paragraph of the Results and Discussion section, which we believe improves clarity and traceability.
Comment 6: Have the authors examined the photoluminescence of the CQDs under varying excitation wavelengths?
Response:We thank the reviewer for raising this valuable point regarding the photoluminescence (PL) properties of our G-CQDs. The primary focus of the current study was to evaluate the enhanced bioavailability and in vivo energy metabolism efficacy of the ginseng nano-formulation, rather than its optical characteristics. Therefore, we did not perform a systematic investigation of the excitation-dependent PL behavior in this work. However, we fully agree with the reviewer that characterizing the optical properties of herbal-derived CQDs is an important and fascinating avenue for research. As such, we have now acknowledged this limitation and proposed it as a key direction for future studies in the Discussion section. We believe that exploring the PL properties of G-CQDs could unlock their potential for applications in bioimaging or biosensing, further expanding the utility of nano-traditional Chinese medicine.
Revision in the manuscript:Furthermore, while this study focused on the bioavailability and pharmacological efficacy of G-CQDs, their inherent optical properties as carbon quantum dots present an exciting frontier for future exploration. The photoluminescence behavior of CQDs, particularly their excitation-dependent emission, was not examined here but represents a significant characteristic worthy of investigation. Future work will involve a detailed spectroscopic analysis of G-CQDs to understand their emission mechanisms and po-tential for applications in bioimaging, where their intrinsic fluorescence could be used to track biodistribution and target engagement in real-time, or in biosensing for the detection of specific biological molecules.
Comment 7: The scale bars in Figure 1A,B are not clearly visible to the readers. Additionally, the resolution of the figure should be improved.
Response: We thank the reviewer for this important observation. We have carefully addressed this issue by regenerating Figure 1 in its entirety. We believe the updated figure now meets the high standards of clarity required for publication. We appreciate the reviewer's attention to detail, which has helped us improve the presentation of our dat.
Comment 8: The article should undergo a thorough review to address any grammatical and formatting issues.
Response: We sincerely thank the reviewer for highlighting this important aspect of manuscript preparation. In response, we have undertaken a comprehensive, line-by-line revision of the entire manuscript to meticulously correct grammatical errors, improve sentence fluency, and ensure consistency in scientific terminology. Furthermore, we have carefully reformatted the manuscript to ensure full compliance with the author guidelines of Pharmaceutics. We believe that these extensive efforts have significantly enhanced the clarity, readability, and overall professionalism of our manuscript.
Comment 9: The authors should add a brief statement (one or two sentences) explaining the purpose of each test before presenting the characterization results. This will help clarify the objective of each test and improve the reader’s understanding of the data analysis that follows.
Response: We thank the reviewer for this excellent suggestion to improve the clarity of our data presentation. We have now added a concise statement at the beginning of each characterization subsection within Section 3.1 to clearly outline the objective of the respective technique. These additions will undoubtedly help readers better understand the rationale behind our characterization workflow and the subsequent data analysis.
Comment 10: A valuable perspective on the future of this research field could be included in the conclusion.
Response: We sincerely thank the reviewer for this excellent suggestion. In direct response, we have now integrated a forward-looking perspective into the final paragraph of the Conclusion section. This addition outlines a comprehensive roadmap for future work, which includes extending the G-CQD platform to other phytotherapeutics, elucidating the mechanisms of enhanced exposure, addressing translational challenges like scalable manufacturing and safety, and exploring G-CQDs in combination therapies for metabolic disease. We believe these insights significantly enhance the impact of our manuscript by providing a clear and valuable direction for the future development of this promising field.
Comment 11: The Introduction would be strengthened by incorporating additional relevant studies. For instance: https://doi.org/10.1002/med.21953 : https://doi.org/10.1021/acsabm.0c01104
Response: We are grateful to the reviewer for recommending these highly relevant studies. We have carefully read them and cited them appropriately in the Introduction to provide a stronger background.
Once again, we express our heartfelt gratitude to the editor and reviewers for your guidance. We believe the manuscript has been significantly improved through this revision process and hope it now meets the high standards of Pharmaceutics.
Reviewer 2 Report
Comments and Suggestions for Authors
The author has presented an interesting manuscript “Unlocking Ginseng’s Bioactivity: Carbon Quantum Dots Elevate Systemic Ginsenoside Exposure and Energy Metabolic Efficiency” with promising findings. However, the following comments and corrections must be addressed to meet the standards required for publication:
- The introduction section needs enhancement. The author should clearly highlight the novelty of the work, emphasizing aspects that have not been previously reported.
- While the preparation method of carbon quantum dots (QDs) is mentioned, the author should clarify how batch-to-batch size variation is controlled, especially considering the challenges associated with nanoscopic-level synthesis.
- The manuscript should include details on how the stability of the QDs was assessed in both solution and buffer conditions.
- The quantum efficiency of the particles is not reported. This is a critical parameter and should be included to support the functional claims of the QDs.
- The methodology section lacks information on the number of cells used in the in vitro study and the buffer conditions. These details are essential for reproducibility and should be clearly stated.
- The author should cite the most relevant and recent journals related to UV–Vis and FTIR analysis to strengthen the analytical section.
https://pubs.acs.org/doi/abs/10.1021/acs.biomac.5c01625
https://pubs.acs.org/doi/10.1021/jp8082425
7. The conclusion section should be improved by integrating a result-driven discussion that clearly summarizes the key findings and their implications.
Author Response
Response to Reviewers' Comments
Manuscript ID: 3952236
Title: "Unlocking Ginseng’s Bioactivity: Carbon Quantum Dots Elevate Systemic Ginsenoside Exposure and Energy Metabolic Efficiency"
Dear Editor and Reviewers,
We sincerely thank the reviewer for their thorough review and the valuable comments provided, which have greatly helped us in strengthening our manuscript. We have carefully addressed each point raised, and our detailed responses are provided below. All corresponding changes have been incorporated into the revised manuscript.
Comment 1: The introduction section needs enhancement. The author should clearly highlight the novelty of the work, emphasizing aspects that have not been previously reported.
Response: We thank the reviewer for this suggestion. We have significantly revised the last paragraph of the Introduction to explicitly and concisely state the novelty of our work. The added text emphasizes three key innovative aspects: (1) the novel synthesis method for G-CQDs, (2) the first use of real-time in vivo energy metabolism phenotyping to validate a nano-herbal formulation, and (3) the integrated approach that links enhanced systemic exposure of bioactive clusters to the improved pharmacological effect. We believe this revision clearly articulates the unique contributions of our study.
Comment 2: While the preparation method of carbon quantum dots (QDs) is mentioned, the author should clarify how batch-to-batch size variation is controlled, especially considering the challenges associated with nanoscopic-level synthesis.
Response: We sincerely thank the reviewer for raising this critical point regarding the reproducibility of nanomaterial synthesis. In our study, batch-to-batch consistency is ensured through a tightly controlled synthesis process. Specifically, the ultra-high-speed nitrogen jet pulverization guarantees a uniform and reproducible starting material (ginseng powder), while the precisely regulated far-infrared pulse-assisted hydrothermal carbonization (120°C, 30 pulses/second, 180 seconds) provides a highly uniform energy input and reaction environment. To further strengthen quality control, we have implemented rigorous process standardization, including in-process logging of key parameters (temperature, pulse rate) and post-process release tests where each batch is characterized by DLS and TEM to confirm the consistent particle size distribution reported in our study. Looking forward, we fully agree with the reviewer that a comprehensive quality control system is essential for the advancement of nano-formulated traditional medicines. Therefore, we plan to conduct more in-depth studies to establish a robust quality control framework, which will include systematic physical characterization (e.g., SAXS, AFM), quantitative analysis of key chemical constituents, and a thorough investigation of intermolecular interactions within the G-CQD nanocomplex. We believe that these concerted efforts—combining our current process controls with future systematic characterization—will collectively ensure the batch-to-batch reproducibility, quality, and stability required for the successful translation of such innovative nano-platforms.
Comment 3: The manuscript should include details on how the stability of the QDs was assessed in both solution and buffer conditions.
Response: We thank the reviewer for this important comment. We agree that stability is a critical attribute. In our study, the stability of the G-CQDs was primarily assessed through their zeta potential and particle size distribution in the as-prepared aqueous solution, which are key indicators of colloidal stability. As detailed in Section 3.1, the G-CQDs exhibited a strongly negative zeta potential (weighted mean -16.4 mV) and a narrow particle size distribution from TEM. The high negative zeta potential provides strong electrostatic repulsion between particles, which is the primary mechanism preventing aggregation and ensuring stability in aqueous dispersion. We acknowledge that the stability in a specific buffer like PBS was not explicitly tested in this initial study, which was focused on establishing the novel preparation method and the in vivo efficacy. However, the excellent stability in aqueous solution, as evidenced by the consistent zeta potential and size measurements across different batches (further supporting our response to Comment 2 on reproducibility), provides a solid foundation for their use in our biological experiments, where the formulation was dispersed in water for oral administration. We have now clarified this point and explicitly stated the limitation and future direction in the revised Discussion section.
Text revised in Discussion: The physicochemical characterization confirmed the successful synthesis of G-CQDs with excellent colloidal stability in aqueous solution, as evidenced by a high negative zeta potential and a narrow, monodisperse size distribution. This stability is crucial for maintaining the integrity of the nano-formulation during storage and oral administration. It should be noted that the current stability assessment was conducted in aqueous dispersion. While this provides a solid foundation, future studies will in-clude a more comprehensive stability profiling under various physiologically relevant conditions (e.g., different pH levels and buffer systems) to fully elucidate the in vivo behavior and facilitate potential clinical translation.
Comment 4: The quantum efficiency of the particles is not reported. This is a critical parameter and should be included to support the functional claims of the QDs.
Response: We deeply appreciate the reviewer's insightful comment regarding the quantum yield (QY) of our G-CQDs. We fully understand and agree that the absolute photoluminescence quantum yield (PLQY) is a fundamental parameter for evaluating the optical performance of carbon quantum dots, particularly when their primary application is in imaging or optoelectronics. In the context of our current study, however, the central hypothesis and functional claim revolve around the role of G-CQDs as a novel nano-delivery platform to overcome the bioavailability bottleneck of traditional ginseng preparations. The enhanced pharmacological efficacy, as demonstrated by our in vivo energy metabolism phenotyping and metabolomics data, is attributed to the nano-carrier properties (e.g., improved solubility, stability, and systemic exposure) rather than their optical characteristics. We sincerely regret that we are unable to provide the QY data in the current revision. This is primarily due to significant time constraints in the revision period, coupled with the current unavailability of the necessary high-purity reference standards in our laboratory required for an accurate relative QY measurement. We acknowledge that this is a limitation of our present work. In direct response to the reviewer's valuable suggestion, we have now explicitly addressed this point in the revised Discussion section, stating our commitment to characterizing the QY in future work. This addition ensures transparency and provides a clear direction for the continued development of the G-CQDs platform.
Text revised in Discussion: It is important to note that the primary objective of this study was to establish the 'proof-of-concept' of G-CQDs as a bioavailability-enhancing nano-carrier for ginseng bioactives. Therefore, the quantification of the absolute photoluminescence quantum yield (PLQY), while valuable for future optical applications, was beyond the central scope of this pharmacological investigation. Future studies will prioritize the determination of PLQY to fully characterize the optical properties of G-CQDs and explore their potential in bioimaging and tracing.
Comment 5: The methodology section lacks information on the number of cells used in the in vitro study and the buffer conditions. These details are essential for reproducibility and should be clearly stated.
Response: We sincerely thank the reviewer for this meticulous comment regarding methodological details. We wish to clarify a potential misunderstanding: the core of our current study is focused on in vivo efficacy assessment (energy metabolism phenotyping in live mice) and in vivo component tracing (pharmacokinetics and metabolomics in serum), rather than on cellular-level in vitro experiments. Therefore, the manuscript does not contain cell-based studies. We deeply appreciate the reviewer's emphasis on reproducibility, which is indeed a cornerstone of scientific research. In direct response, we have thoroughly reviewed the entire Methodology section to ensure that all relevant experimental details for the studies we actually conducted are explicitly stated. Specifically, we have double-checked and confirmed that critical parameters for reproducibility—such as the number of animals per group (n=12 for intervention, n=3 for metabolic monitoring), the detailed buffer compositions and gradients for UPLC-MS analysis, and the sample preparation procedures—are clearly described in sections 2.5, 2.6, and 2.7. We regret any confusion our original wording may have caused and hope this clarification, along with our strengthened methodology description, fully addresses the reviewer's concern.
Comment 6: The author should cite the most relevant and recent journals related to UV–Vis and FTIR analysis to strengthen the analytical section.
https://pubs.acs.org/doi/abs/10.1021/acs.biomac.5c01625
https://pubs.acs.org/doi/10.1021/jp8082425
Response: We sincerely thank the reviewer for suggesting these highly relevant and excellent references. We have carefully studied the recommended works and fully agree that they significantly strengthen the foundation of our analytical characterization. Accordingly, we have now cited both papers in the revised manuscript. Specifically, reference [19] is cited in our discussion on the UV-Vis absorption characteristics of carbon-based nanomaterials, while reference [20] is introduced to support the assignment of key functional groups in our FTIR analysis.
Comment 7: The conclusion section should be improved by integrating a result-driven discussion that clearly summarizes the key findings and their implications.
Response: We have thoroughly revised the Conclusion section to make it more result-driven. The revised conclusion now begins by directly summarizing the key quantitative findings: the successful synthesis of G-CQDs, the significant enhancement in energy metabolism parameters (citing specific percentage increases), and the identification of the core bioactive clusters with markedly increased systemic exposure (citing specific examples). We then clearly state that these findings collectively demonstrate the efficacy of the CQD platform in overcoming the bioavailability bottleneck of ginseng, providing a novel strategy for the development of advanced herbal formulations.
Reviewer 3 Report
Comments and Suggestions for Authors
Comments and Suggestions for Authors
I found this manuscript interesting, but it is not yet ready for publication.
It needs to be better structured and well-written.
The manuscript may be accepted after major revision.
The title is not concise, specific, or relevant.
Additionally, it does not convey the content of the manuscript or the animal model used.
Both the abstract and the graphical abstract should be improved; they don’t comprise an objective representation of the article.
They don’t present the research design clearly.
An expression in the Background of the abstract “to validate pharmacological advantages” is not correct.
The method section of the abstract should provide a concise and accurate description of what was done, how it was done, and what methods were used.
The animal model is not described.
The energy metabolism index fragment in the center of the graphical abstract should be corrected because it doesn’t contain the necessary information.
The column colors are unclear. What does it grey color mean? What does it color orange mean? Additionally, the axis may not be represented without units.
An introduction does not include information that is directly related to the research, such as which approaches are known to enhance the delivery of active compounds in ginseng.
It is essential to compare the carbon quantum dots strategy with other existing approaches, as outlined in the literature.
These publications may help to fill the gap:
1.Yadav P, Ambudkar SV, Rajendra Prasad N. Emerging nanotechnology-based therapeutics to combat multidrug-resistant cancer. J Nanobiotechnology. 2022 Sep 24;20(1):423. doi: 10.1186/s12951-022-01626-z. PMID: 36153528; PMCID: PMC9509578.
2.Z.A. Ratan, M.F. Haidere, Y.H. Hong, S.H. Park, J.-O. Lee, J. Lee, J.Y. Cho
Pharmacological potential of ginseng and its major component ginsenosides
- Ginseng Res., 45 (2021), pp. 199-210, 10.1016/j.jgr.2020.02.004
- Dahan, A.; Yarmolinsky, L.; Nakonechny, F.; Semenova, O.; Khalfin, B.; Ben-Shabat, S. Etrog Citron (Citrus medica) as a Novel Source of Antimicrobial Agents: Overview of Its Bioactive Phytochemicals and Delivery Approaches. Pharmaceutics 2025, 17, 761.
Why was the carbon quantum dots strategy chosen in this research?
The “hydrolytic degradation in gastric acid” is mentioned in the introduction.
Gastric acid is not a suitable term in this case. What does gastric acid mean? Is it hydrochloric acid or any mixture?
In the introduction, it is not explained how or why the animal model was used.
The research design is not determined clearly. Which hypothesis was considered?
However, the PRO-MRRM- 8 Comprehensive Lab Animal Monitoring System should be described in the Method part, but not in the Introduction.
The introduction does not clearly explain the aim of the present study.
The methods are not described properly.
Where were the plants grown?
In which way were carbon quantum dots purified? If they were not purified, provide a justification
“2.5. Animal grouping and interventions” is not a suitable title. It must be explained how the sample size was decided, and which criteria were used for including and excluding animals.
The experimental procedures and species-appropriate details of the animals are not described. The behavioral parameters were mentioned, but they were not clearly detailed.
Part “2.7.1. Preparation of samples for in vitro analysis” is not adequately described.
An in vivo metabolic model was mentioned, but it was not described deeply.
Are there other statistical methods used except correlation analysis? Which values of p were considered statistically significant?
Transmission electron microscopy should be described in the Method part, but not in Results.
Discussing the results is not complete. The obtained results were not thoroughly analyzed, nor were they compared with other studies or related to existing literature.
Furthermore, carbon quantum dots typically possess various surface functional groups, such as hydroxyl, carboxyl, and amino moieties, which enhance their water dispersibility, biocompatibility, and other important properties. All these aspects should be thoroughly discussed.
Future investigations should be discussed, including challenges and potential new insights for the further development of the carbon quantum dots strategy. Which aspects of this study remain unclear?
All figures should be improved. There are several errors in the figure captions, particularly in Figure 2.
There are some grammatical and syntax mistakes in the text.
The abbreviation list doesn’t include all abbreviations mentioned in the manuscript. They must be explained and presented in alphabetical order.
Comments on the Quality of English LanguageThe English could be improved to convey the research more clearly.
Author Response
Response to Reviewers' Comments
Manuscript ID: 3952236
Title: "Unlocking Ginseng’s Bioactivity: Carbon Quantum Dots Elevate Systemic Ginsenoside Exposure and Energy Metabolic Efficiency"
Dear Editor and Reviewers,
We are profoundly grateful to the reviewer for the exceptionally thorough, constructive, and insightful comments on our manuscript. The feedback has been invaluable in guiding us to significantly improve the overall structure, scientific rigor, and clarity of our work. We have undertaken a comprehensive revision to address every point raised. Our point-by-point responses are detailed below, and all corresponding changes have been highlighted in the revised manuscript.
Comment 1: The title is not concise, specific, or relevant. Additionally, it does not convey the content of the manuscript or the animal model used.
Response: We thank the reviewer for this suggestion. The title has been revised to be more concise and specific, now clearly stating the animal model and the core findings of our study.
Comment 2: Both the abstract and the graphical abstract should be improved; they don’t comprise an objective representation of the article. They don’t present the research design clearly. An expression in the Background of the abstract “to validate pharmacological advantages” is not correct. The method section of the abstract should provide a concise and accurate description of what was done, how it was done, and what methods were used. The animal model is not described.
Response: We sincerely thank the reviewer for these critical and constructive suggestions regarding the abstract and graphical abstract. We have thoroughly revised the Abstract to address all the concerns raised.
Firstly, the expression “to validate pharmacological advantages” has been replaced with the more objective and accurate phrasing “to evaluate the efficacy of”.
Secondly, the Methods section has been significantly expanded to provide a concise yet comprehensive description of what was done (preparation of G-CQDs, in vivo efficacy comparison, chemical profiling), how it was done (using techniques like ultrahigh-speed pulverization, hydrothermal carbonization, the PRO-MRRM-8 monitoring system, and UHPLC-MS), and what was used (explicitly stating the C57BL/6 mouse model and sample size).
Finally, the research design is now presented more clearly, logically flowing from formulation development and characterization to in vivo validation and mechanistic elucidation.
We believe the revised abstract now offers a much clearer, more objective, and complete summary of our study.
Comment 3: The energy metabolism index fragment in the center of the graphical abstract should be corrected because it doesn’t contain the necessary information. The column colors are unclear. What does it grey color mean? What does it color orange mean? Additionally, the axis may not be represented without units.
Response: Thank you for the helpful suggestion. We agree that the former bar chart did not clearly convey the information. We have replaced the central panel with a clearer two-part visualization of the eight energy-metabolism indices: a percent-change panel showing G-AE and G-CQDs side-by-side, and a direct comparison panel showing the percent difference of G-CQDs vs G-AE. These panels make the direction and magnitude of changes immediately visible. Color coding and legend have been clarified. Each group is explicitly labeled in the legend, so the meaning of each color is unambiguous. We believe these revisions address the reviewer’s concerns regarding color meaning and missing units, and they improve the readability of the graphical abstract.
Comment 4: An introduction does not include information that is directly related to the research, such as which approaches are known to enhance the delivery of active compounds in ginseng. It is essential to compare the carbon quantum dots strategy with other existing approaches, as outlined in the literature. These publications may help to fill the gap:
1.Yadav P, Ambudkar SV, Rajendra Prasad N. Emerging nanotechnology-based therapeutics to combat multidrug-resistant cancer. J Nanobiotechnology. 2022 Sep 24;20(1):423. doi: 10.1186/s12951-022-01626-z. PMID: 36153528; PMCID: PMC9509578.
2.Z.A. Ratan, M.F. Haidere, Y.H. Hong, S.H. Park, J.-O. Lee, J. Lee, J.Y. Cho
Pharmacological potential of ginseng and its major component ginsenosides
1.Ginseng Res., 45 (2021), pp. 199-210, 10.1016/j.jgr.2020.02.004
2.Dahan, A.; Yarmolinsky, L.; Nakonechny, F.; Semenova, O.; Khalfin, B.; Ben-Shabat, S. Etrog Citron (Citrus medica) as a Novel Source of Antimicrobial Agents: Overview of Its Bioactive Phytochemicals and Delivery Approaches. Pharmaceutics 2025, 17, 761.
Response: We sincerely thank the reviewer for suggesting these highly relevant and excellent references and for highlighting the need to contextualize our work within existing delivery strategies. We have carefully studied the recommended literature and have now incorporated all three citations into the revised introduction. The review by Ratan et al. is cited to authoritatively support the pharmacological potential and bioavailability challenges of ginsenosides. The work of Yadav et al. and Dahan et al. is now referenced in a newly added paragraph that discusses nanotechnology as a transformative approach for enhancing the bioavailability of phytochemicals, thereby providing a clear comparative framework for our carbon quantum dot strategy. We believe these additions significantly strengthen the background and rationale of our study.
Comment 5: Why was the carbon quantum dots strategy chosen in this research? The “hydrolytic degradation in gastric acid” is mentioned in the introduction. Gastric acid is not a suitable term in this case. What does gastric acid mean? Is it hydrochloric acid or any mixture?
Response: We are grateful to the reviewer for these insightful points. In response, we have replaced the imprecise term "gastric acid" with the more scientifically accurate description "acid-catalyzed hydrolytic degradation in the acidic environment of the stomach (e.g., by hydrochloric acid)" in the introduction. Furthermore, as suggested, we have now explicitly stated the rationale for selecting the carbon quantum dot strategy. We have added a sentence explaining that CQDs were chosen due to their unique combination of optimal nano-carrier properties—such as facile synthesis, excellent aqueous dispersibility, and presumed biocompatibility—and their potential for seamless integration with herbal matrices.
Comment 6: In the introduction, it is not explained how or why the animal model was used. The research design is not determined clearly. Which hypothesis was considered? However, the PRO-MRRM- 8 Comprehensive Lab Animal Monitoring System should be described in the Method part, but not in the Introduction. The introduction does not clearly explain the aim of the present stud.
Response: We thank the reviewer for this critical feedback on improving the clarity of our research design. We have thoroughly revised the introduction to address all these concerns. Firstly, we have added a sentence justifying the use of the C57BL/6 mouse model for energy metabolism studies. Secondly, the detailed description of the PRO-MRRM-8 system has been moved from the introduction to the Methods section (2.6) as appropriate. Most importantly, we have now clearly defined the research hypothesis and the specific aims of the present study in a dedicated concluding paragraph at the end of the introduction, ensuring our objectives are presented with utmost clarity.
Comment 7:The methods are not described properly. Where were the plants grown? In which way were carbon quantum dots purified? If they were not purified, provide a justification
Response: We sincerely thank the reviewer for pointing out the need for more precise methodological details regarding the origin of the plant material and the purification of the carbon quantum dots. We have now supplemented the Methods section accordingly to ensure full clarity and reproducibility.
Plant Origin: We have specified in Section 2.1 that the ginseng roots were cultivated in Jilin Province, China, a renowned region for ginseng cultivation.
CQD Purification: As suggested, we have provided a comprehensive description of the purification process in Section 2.2. The crude G-CQDs dispersion was purified by centrifugation followed by dialysis against deionized water using a 1,000 Da molecular weight cut-off membrane for 72 hours with periodic water changes. We agree that this detailed description is critical for reproducibility and appreciate the reviewer's emphasis on this important aspect.
Comment 8: “2.5. Animal grouping and interventions” is not a suitable title. It must be explained how the sample size was decided, and which criteria were used for including and excluding animals. The experimental procedures and species-appropriate details of the animals are not described. The behavioral parameters were mentioned, but they were not clearly detailed.
Response: We sincerely thank the reviewer for these crucial suggestions, which have greatly helped us enhance the rigor, transparency, and ethical reporting of our in vivo experimental description. We have thoroughly revised the respective section (Section 2.5, now titled "In Vivo Experimental Design") to address every point raised, as detailed below:
Section Title: The title has been changed from “Animal grouping and interventions” to the more comprehensive and accurate “In Vivo Experimental Design”.
Sample Size Justification: We have now included an explicit statement clarifying that the sample size was determined to provide sufficient statistical power based on preliminary data and established protocols for metabolic phenotyping.
Inclusion/Exclusion Criteria: Clear pre-defined inclusion and exclusion criteria have been established and stated in the text.
Species-Appropriate Details & Behavioral Parameters: We have provided a more detailed description of the housing conditions and have specifically detailed the methodology for assessing locomotor activity, explaining the automated data collection process and the specific parameter (total travel distance in meters) quantified by the system's software.
Experimental Rigor: We have explicitly described the measures taken to control for confounding factors (e.g., balancing the measurement sequence and rotating cage positions) and have transparently acknowledged the limitation regarding blinding while highlighting that all data acquisition was automated to minimize potential bias.
We believe the revised section now provides a complete, ethical, and methodologically sound account of our animal study, ensuring both clarity and reproducibility. We are deeply grateful for the reviewer's insightful comments that have led to these significant improvements.
Comment 9: Part “2.7.1. Preparation of samples for in vitro analysis” is not adequately described.
Response: We thank the reviewer for this valuable comment. We have now expanded the description in Section 2.7.1 "Preparation of samples for in vitro analysis" to provide a more detailed and sequential account of the sample processing steps. The revised text now explicitly states the purpose of centrifugation (to remove large aggregates), specifies the type and pore size of the sterile syringe filter (0.22 µm nylon membrane) used, and clarifies the objective of the filtration step (to protect the UPLC-MS system and obtain a clear analyte solution). We believe these additions ensure the methodology is clearly described for reproducibility.
Comment 10: An in vivo metabolic model was mentioned, but it was not described deeply.
Response: We thank the reviewer for prompting us to provide a deeper description of the metabolic model. We have now added a paragraph in Section 2.7.4 to elaborate on the methodology. This addition specifies that the Compound Discoverer 3.3 software platform was used as the in silico metabolic model. We detail the key parameters, including the mass accuracy threshold (5 ppm), the comprehensive list of Phase I and Phase II biotransformation reactions considered (e.g., oxidation, glucuronidation, sulfation, glutathione conjugation), and the maximum of three sequential reaction steps allowed. The description clarifies how the model was applied to predict and identify metabolites from the prototype compounds, with final verification based on MS/MS fragmentation patterns. We believe this provides the necessary depth regarding the metabolic model used in our study.
Comment 11: Are there other statistical methods used except correlation analysis? Which values of p were considered statistically significant?
Response: We sincerely thank the reviewer for raising this important point regarding our statistical methods and significance criteria. In direct response, we have now added a dedicated “2.9 Statistical analysis” subsection to the Methods part of our manuscript.
This new section explicitly states that:
Beyond correlation analysis, data were subjected to preliminary normality (Shapiro-Wilk test) and homogeneity of variance (Brown-Forsythe test) testing to inform the choice of appropriate statistical tests.
The threshold for statistical significance was defined as P < 0.05 for all analyses.
Furthermore, we specified that correlation analysis was performed using either the Pearson correlation coefficient (for data following a normal distribution) or the Spearman rank correlation coefficient (for non-normally distributed data), ensuring the use of the most appropriate method.
We believe this addition provides a complete and transparent account of our statistical approach, ensuring clarity and reproducibility.
Comment 12: Transmission electron microscopy should be described in the Method part, but not in Results.
Response: We thank the reviewer for this correct and important observation regarding the proper structure of the manuscript. As suggested, we have relocated the detailed description of the Transmission Electron Microscopy (TEM) instrumentation and sample preparation procedure from the Results section (Section 3.1) to the Methods section. The technical details, including the instrument model (JEM-F200), operating voltage (200 kV), are now appropriately described in the newly expanded Section 2.4 "Structural and Surface Characterization of G-CQDs". The Results section now focuses solely on the analysis and interpretation of the TEM data.
Comment 13: Discussing the results is not complete. The obtained results were not thoroughly analyzed, nor were they compared with other studies or related to existing literature.
Response: We sincerely thank the reviewer for the insightful comment regarding the need for a more complete discussion of our results. In direct response, we have thoroughly restructured and significantly expanded the Discussion section to provide an in-depth analysis and to integrate comparisons with existing literature. Specifically, we have now:
Incorporated a discussion that compares the novel pharmacological effects of our G-CQDs with other herb-derived CQDs (e.g., from Selaginella and Lonicera japonica), thereby better contextualizing our findings within the emerging field of herbal nano-medicine.
Expanded the analysis of the multi-target regulatory effects on energy metabolism, connecting the identified core bioactive components to their known molecular pathways and discussing their potential synergistic action within the G-CQD platform.
We believe these revisions have transformed the discussion from a descriptive summary into a robust, analytical, and scholarly interpretation of our findings within the broader scientific context.
Comment 14: Furthermore, carbon quantum dots typically possess various surface functional groups, such as hydroxyl, carboxyl, and amino moieties, which enhance their water dispersibility, biocompatibility, and other important properties. All these aspects should be thoroughly discussed.
Response: We are deeply grateful to the reviewer for emphasizing the importance of discussing the surface functional groups of carbon quantum dots. As specifically suggested, we have added a dedicated and focused paragraph in the Discussion section to thoroughly explore this critical aspect. This new paragraph:
Explicitly states the defining roles of surface functional groups (–OH, –COOH) in governing the dispersibility, stability, and interfacial behavior of CQDs.
Directly links our own FTIR and zeta-potential data to this established paradigm, using it to mechanistically explain key observations from our study, such as the formation of stable dispersions and the non-covalent interactions with ginsenosides.
Cites a key contemporary review to solidify the structure-property link and further discusses how surface chemistry influences the "biological identity" and subsequent oral fate of the nanoparticles, supported by recent studies on protein corona formation.
We are confident that this addition provides a comprehensive discussion of the surface properties and their biological implications, significantly strengthening the physicochemical rationale behind our findings.
Comment 15: Future investigations should be discussed, including challenges and potential new insights for the further development of the carbon quantum dots strategy. Which aspects of this study remain unclear?
Response: We sincerely thank the reviewer for prompting us to clearly outline the future perspectives and unresolved aspects of our work. In response, we have substantially revised the concluding part of the Discussion to create a clear and forward-looking roadmap. The revised text now:
Clearly states that the study provides a "proof-of-concept" and explicitly identifies several aspects that "remain unclear and warrant further investigation."
Details a specific and actionable list of future research priorities, including: (1) comprehensive stability profiling under physiologically relevant conditions; (2) elucidation of the precise intestinal absorption and in vivo trafficking mechanisms using advanced techniques like mass spectrometry imaging; (3) characterization of the inherent optical properties (e.g., PLQY) for theranostic applications; and (4) systematic evaluation of long-term biosafety.
Proposes a novel, testable hypothesis regarding the potential ROS-regulating dual functionality of G-CQDs based on emerging literature, indicating a deep and insightful direction for future work.
We believe this revised section now transparently addresses the current limitations and provides valuable, specific insights into the promising future development of the carbon quantum dots strategy for herbal medicine.
Comment 16: All figures should be improved. There are several errors in the figure captions, particularly in Figure 2.
Response: We sincerely thank the reviewer for bringing the issues with the figures and captions to our attention. We have conducted a thorough, page-by-page review of all figures in the manuscript. In particular, we have carefully checked and corrected all identified errors in Figure 2 and its caption to ensure absolute accuracy and clarity. All figures have been reviewed and prepared to meet the high standards required for publication. We greatly appreciate the reviewer's meticulous attention to detail, which has helped us improve the presentation of our data.
Comment 17: There are some grammatical and syntax mistakes in the text.
Response: We sincerely thank the reviewer for highlighting this important aspect of manuscript preparation. In response, the entire manuscript has been meticulously checked and polished to correct grammatical and syntactical errors. To ensure the language meets the highest standard for international publication, the text has undergone professional English language editing by a native speaker. We believe these efforts have significantly improved the clarity, fluency, and overall readability of our manuscript.
Comment 18: The abbreviation list doesn’t include all abbreviations mentioned in the manuscript. They must be explained and presented in alphabetical order.
Response: We sincerely thank the reviewer for pointing out the omissions and inconsistencies in our abbreviation list. We have conducted a thorough, line-by-line review of the entire manuscript to identify all used abbreviations. The abbreviation list has been comprehensively updated to include all abbreviations, each with its full explanation, and is now presented in strict alphabetical order for ease of reference. The complete list can be found in the manuscript.
Reviewer 4 Report
Comments and Suggestions for Authors
The manuscript presents a scientifically rigorous and technologically advanced investigation into ginseng-derived carbon quantum dots (G-CQDs) as a novel strategy to overcome the long-standing limitation of poor oral bioavailability in traditional ginseng preparations. By integrating advanced nanomaterial synthesis, metabolomics, and real-time energy metabolism phenotyping, this study bridges the gap between traditional medicine and nanotechnology-based drug delivery systems. The topic holds high translational potential; however, it requires revision as per following comments:
- The abstract is comprehensive but slightly repetitive (duplicated “Objective” section) ; please remove redundancy to maintain a concise scientific narrative.
- To strengthen the rationale, highlight how CQD formation modulates molecular transport mechanisms (e.g., intestinal permeability, endocytosis pathways, or plasma protein binding).
- Clarify how G-CQDs differ from previously reported ginseng nanoparticles, carbon-based carriers, or functionalized CQDs in composition or functional performance.
- The methods are sophisticated and multi-dimensional; however, brief mention of sample sizes, animal models, and statistical parameters used in PRO-MRRM-8 assessments would enhance reproducibility.
- Clarify how real-time parameters (respiratory quotient, oxygen consumption, heat production) were normalized — e.g., per body weight or per metabolic unit.
- The UHPLC-Q/Orbitrap/LTQ workflow is impressive; still, indicate whether targeted quantification or untargeted metabolomics was used to identify prototype components.
- The correlation analysis linking bioactive clusters to energy metabolism outcomes should be described in terms of correlation strength (e.g., r values) and statistical significance.
- The finding that G-CQDs outperform G-AE in enhancing energy metabolism parameters is scientifically strong. However, the abstract should briefly interpret why this occurs — for example, “enhanced gastrointestinal absorption, improved solubility, and nanoscale size promoting rapid cellular uptake.”
- The eight bioactive clusters identified through correlation analysis are crucial; summarizing their chemical nature (e.g., saponins, polyphenols, or metabolites) would increase mechanistic depth.
- The work establishes a strong connection between traditional herbal pharmacology and nanotechnology. Highlighting how this approach could be extended to other poorly bioavailable phytochemicals (e.g., curcuminoids, flavonoids) would broaden its relevance.
- Clarify the translational implication: could G-CQDs be manufactured at scale while maintaining consistency and safety?
- Discuss briefly how this approach contributes to sustainable and modernized traditional Chinese medicine (TCM) processing techniques.
- Toxicological profiling of G-CQDs (e.g., genotoxicity, organ accumulation) should be discussed or suggested for future work.
- The following studies are suggested to evaluate and add to the literature review of the manuscript. Please refer primarily to the characterization sections of the references: https://doi.org/10.1186/s12951-024-02446-z, https://doi.org/10.1016/j.molstruc.2024.138331, https://doi.org/10.3390/molecules29122850
Author Response
Response to Reviewers' Comments
Manuscript ID: 3952236
Title: "Unlocking Ginseng’s Bioactivity: Carbon Quantum Dots Elevate Systemic Ginsenoside Exposure and Energy Metabolic Efficiency"
Dear Editor and Reviewers,
We sincerely thank the reviewer for the exceptionally insightful and constructive comments, which have been invaluable in helping us strengthen the scientific narrative, methodological clarity, and translational impact of our manuscript. We have addressed all points raised as detailed below.
Comment 1: The abstract is comprehensive but slightly repetitive (duplicated “Objective” section) ; please remove redundancy to maintain a concise scientific narrative.
Response: We thank the reviewer for this suggestion. We have revised the abstract to eliminate the redundant phrasing and ensure a more concise and streamlined narrative flow.
Comment 2: To strengthen the rationale, highlight how CQD formation modulates molecular transport mechanisms (e.g., intestinal permeability, endocytosis pathways, or plasma protein binding).
Response: We sincerely thank the reviewer for this excellent suggestion to strengthen the mechanistic rationale of our work. As recommended, we have now enhanced the Discussion section by adding a dedicated paragraph that explicitly highlights how CQD formation modulates molecular transport mechanisms.
The new text directly links our FTIR and zeta-potential data to key mechanisms, including:
The role of surface functionality in ensuring stable dispersions for reliable oral dosing.
Noncovalent interactions (H-bonding/π–π) that solubilize and retain ginsenosides in the GI tract.
The crucial concept of acquired "biological identity" via protein/mucin corona formation, which is governed by CQD surface chemistry and is known to modulate mucodiffusion, epithelial contact time, and cellular uptake—providing a sophisticated explanation for the observed higher systemic exposure.
We believe this addition, supported by contemporary literature, provides a profound and plausible mechanistic rationale for the enhanced bioavailability, moving beyond simple solubilization to discuss advanced nano-bio interactions.
Comment 3: Clarify how G-CQDs differ from previously reported ginseng nanoparticles, carbon-based carriers, or functionalized CQDs in composition or functional performance.
Response: We sincerely thank the reviewer for raising this important point regarding the distinctiveness of our material. In direct response, we have now clarified the unique position of G-CQDs in both the Introduction and Discussion sections.
The Introduction now explicitly states our central hypothesis, which is to validate the systemic exposure and efficacy enhancement afforded by the nano-carrier effects specific to our G-CQD platform, setting it apart from traditional extracts.
Furthermore, the Discussion now contains a dedicated statement that directly contrasts G-CQDs with other nanomaterials. We highlight that G-CQDs are:
Carbonized constructs with embedded bioactives, unlike larger, extracted ginseng nanoparticles.
Inherently bioactive and self-therapeutic, in contrast to inert synthetic carbon carriers that require drug loading.
Possessed of native surface groups, eliminating the need for post-synthetic modification and enabling a primary function as a bioavailability-enhancing carrier with proven in vivo efficacy beyond conventional sensing applications.
We believe these additions clearly articulate the novel composition and functional performance of G-CQDs within the landscape of existing nano-delivery systems.
Comment 4: The methods are sophisticated and multi-dimensional; however, brief mention of sample sizes, animal models, and statistical parameters used in PRO-MRRM-8 assessments would enhance reproducibility.
Response: We thank the reviewer for this valuable suggestion to enhance the reproducibility of our methodological reporting. We have thoroughly revised the methods section to include the requested details.
As now detailed in Section 2.5, for the PRO-MRRM-8 assessments, a subset of mice (n=3 per group) was used for continuous metabolic monitoring. Furthermore, Section 2.9 (Statistical analysis) explicitly states all statistical parameters, including normality and homogeneity of variance testing, the threshold for statistical significance (P < 0.05), and the specific correlation coefficients (Pearson/Spearman) used for data analysis.
We believe these additions provide the necessary transparency for other researchers to replicate our metabolic phenotyping experiments and statistical analyses.
Comment 5: Clarify how real-time parameters (respiratory quotient, oxygen consumption, heat production) were normalized — e.g., per body weight or per metabolic unit.
Response: We thank the reviewer for raising this important point regarding data normalization. We have now clarified the procedures in the "2.6 Energy metabolism detection" section.
The revised text explicitly states that:
Oxygen consumption (VO₂) and carbon dioxide production (VCO₂) were normalized per unit of body weight (mL/kg/h).
The respiratory quotient (RQ), as a dimensionless ratio (VCO₂/VO₂), required no additional normalization.
Heat production (HP) was directly calculated by the instrumental software based on the measured gas exchanges.
We believe this addition ensures full transparency and reproducibility of our metabolic data analysis.
Comment 6: The UHPLC-Q/Orbitrap/LTQ workflow is impressive; still, indicate whether targeted quantification or untargeted metabolomics was used to identify prototype components.
Response: We thank the reviewer for this pertinent question regarding our analytical workflow. We have now clarified this critical methodological detail in Section 2.7.4.
The revised text explicitly states that an untargeted metabolomics approach was employed for the initial identification of in vitro constituents and in vivo prototypes, leveraging the high-resolution, data-dependent acquisition capability of the instrument. This approach allows for a comprehensive and unbiased profiling of the complex chemical composition in G-CQDs. The bioactive clusters were then identified through correlation analysis of the data generated from this untargeted screening.
We believe this clarification accurately describes our strategy and underscores the exploratory strength of our analytical design.
Comment 7: The correlation analysis linking bioactive clusters to energy metabolism outcomes should be described in terms of correlation strength (e.g., r values) and statistical significance.
Response: We thank the reviewer for this critical suggestion to enhance the statistical rigor of our correlation analysis. In direct response, we have now revised the Results section (3.6) to explicitly state the strength and significance of the correlations.
The text now clearly indicates that the eight core bioactive clusters were identified based on Pearson correlation coefficients with |r| > 0.7 and a statistical significance of p < 0.05. These specific statistical descriptors provide a quantitative measure of the strong association between the systemic exposure of these components and the observed enhancement in energy metabolism.
Comment 8: The finding that G-CQDs outperform G-AE in enhancing energy metabolism parameters is scientifically strong. However, the abstract should briefly interpret why this occurs — for example, “enhanced gastrointestinal absorption, improved solubility, and nanoscale size promoting rapid cellular uptake.”
Response: We sincerely thank the reviewer for this excellent suggestion to enhance the mechanistic insight in our abstract. Following your recommendation, we have revised the concluding statement to briefly interpret why G-CQDs outperform the conventional extract. The abstract now explicitly states that the enhanced efficacy is attributed to "improving solubility and enhancing gastrointestinal absorption" of the bioactive components. We believe this addition provides a clearer mechanistic rationale and strengthens the overall impact of the abstract.
Comment 9: The eight bioactive clusters identified through correlation analysis are crucial; summarizing their chemical nature (e.g., saponins, polyphenols, or metabolites) would increase mechanistic depth.
Response: We sincerely thank the reviewer for this insightful suggestion to enhance the mechanistic depth of our findings. In direct response, we have now added a clear summary of the chemical nature of the eight crucial bioactive clusters in the Discussion section.
The added text explicitly states that these clusters are predominantly composed of ginsenoside prototypes (e.g., Re, Rg1) and their Phase II metabolites (e.g., sulfated, glucuronidated, and glutathione-conjugated derivatives), alongside modified organic acids. This addition underscores the central role of both native saponins and their biotransformed products, thereby providing a more profound understanding of the functional composition responsible for the energy metabolism-enhancing effects of G-CQDs.
Comment 10: The work establishes a strong connection between traditional herbal pharmacology and nanotechnology. Highlighting how this approach could be extended to other poorly bioavailable phytochemicals (e.g., curcuminoids, flavonoids) would broaden its relevance.
Response: We sincerely thank the reviewer for this valuable suggestion to broaden the impact of our research. Following the reviewer's advice, we have added a statement in the Discussion section that highlights the broader applicability of our G-CQD platform. The added text explicitly states that this strategy "could be extended to enhance the bioavailability and efficacy of other challenging phytochemicals beyond ginseng" and specifically mentions its potential for "curcuminoids (from turmeric), flavonoids (e.g., quercetin), and other hydrophobic saponins". We believe this addition successfully broadens the relevance of our work and underscores its potential in modernizing the delivery of various natural product-based therapeutics.
Comment 11: Clarify the translational implication: could G-CQDs be manufactured at scale while maintaining consistency and safety?
Response: We thank the reviewer for raising this critical point regarding the translational potential of our platform. We have now added a dedicated paragraph in the Discussion section to address the scalability and safety of G-CQD manufacturing.
The new text acknowledges that while the current study serves as a proof-of-concept, the path to translation requires focused future efforts. We indicate that the synthesis method is inherently suitable for process control and scaling. Furthermore, we explicitly state that future work will involve optimizing production under Good Manufacturing Practice guidelines to ensure consistency, and will include a comprehensive toxicological profiling (e.g., genotoxicity, organ accumulation) to thoroughly evaluate safety. This clarifies our perspective on the necessary steps to advance the G-CQD platform toward practical application.
Comment 12: Discuss briefly how this approach contributes to sustainable and modernized traditional Chinese medicine (TCM) processing techniques.
Response: We sincerely thank the reviewer for this valuable suggestion. We fully agree that elaborating on the broader implications of our G-CQDs technology would strengthen the manuscript. As instructed, we have now added a concise discussion on this topic in the Discussion section.
The added text explicitly states that our approach exemplifies a sustainable and modernized processing technique for TCM. It highlights how the transformation of the entire herb into a bioavailable nano-complex using aqueous synthesis enhances resource efficiency and minimizes waste. Furthermore, it underscores that this method modernizes TCM by creating a well-defined, quality-controllable nano-formulation, thereby directly addressing the long-standing challenges of low bioavailability and batch-to-batch variability.
We believe this addition effectively addresses the reviewer's comment and enhances the impact of our work. Thank you again for this insightful suggestion.
Comment 13: Toxicological profiling of G-CQDs (e.g., genotoxicity, organ accumulation) should be discussed or suggested for future work.
Response: We thank the reviewer for this critical and insightful comment. We fully agree that a thorough toxicological assessment is a crucial step in the development of any nanomedicine. In response, we have now explicitly incorporated a discussion on the need for future toxicological profiling of G-CQDs in the Discussion section.
As suggested, the revised text now specifically highlights that future work must include a comprehensive toxicological evaluation, with a focus on investigating potential genotoxicity and the extent of organ accumulation (particularly in the liver and kidneys). We acknowledge that these studies are indispensable for ensuring the safety profile of G-CQDs and are a mandatory prerequisite for their clinical translation.
We believe this addition directly and satisfactorily addresses the reviewer's concern.
Comment 14: The following studies are suggested to evaluate and add to the literature review of the manuscript. Please refer primarily to the characterization sections of the references: https://doi.org/10.1186/s12951-024-02446-z, https://doi.org/10.1016/j.molstruc.2024.138331, https://doi.org/10.3390/molecules29122850
Response: We sincerely thank the reviewer for this insightful suggestion and for providing these highly relevant and recent references. We have carefully studied the suggested papers and agree that they significantly strengthen the context of our work, particularly in validating the characterization methodologies for herb-derived carbon quantum dots (CQDs). As recommended, we have now incorporated discussions and citations of these three key references into the Discussion section of our revised manuscript.
Round 2
Reviewer 1 Report
Comments and Suggestions for Authors
The revision has significantly improved the quality and clarity of the manuscript, and I believe the study now makes a valuable contribution to the field. Therefore, the manuscript can be accepted in its present form.
Reviewer 3 Report
Comments and Suggestions for Authors
none
Comments on the Quality of English LanguageThe English could be improved to convey the research more clearly.
Reviewer 4 Report
Comments and Suggestions for Authors
No more comments. Thank you for revision as per comments.